# Characterization of tryptophan oxidation affecting D1 degradation by FtsH in the photosystem II quality control of chloroplasts

Yusuke Kato[1,2†], Hiroshi Kuroda[3†], Shin-Ichiro Ozawa[1,3†], Keisuke Saito[4], Vivek Dogra[5,6], Martin Scholz[7], Guoxian Zhang[1], Catherine de Vitry[8], Hiroshi Ishikita[4], Chanhong Kim[5], Michael Hippler[1,7], Yuichiro Takahashi[3], Wataru Sakamoto[1]*

[1]Institute of Plant Science and Resources (IPSR), Okayama University, Okayama, Japan; [2]Faculty of Agriculture, Setsunan University, Osaka, Japan; [3]Research Institute for Interdisciplinary Science, Okayama University, Okayama, Japan; [4]Research Center for Advanced Science and Technology, The University of Tokyo, Tokyo, Japan; [5]Shanghai Center for Plant Stress Biology, Center for Excellence in Molecular Plant Sciences, Chinese Academy of Sciences, Shanghai, China; [6]Biotechnology Division, CSIR-Institute of Himalayan Bioresource Technology, Palampur, India; [7]Institute of Plant Biology and Biotechnology, University of Münster, Münster, Germany; [8]Institut de Biologie Physico-Chimique, Unité Mixte de Recherche 7141, Centre National de la Recherche Scientifique and Sorbonne Université Pierre et Marie Curie, Paris, France

*For correspondence:
saka@okayama-u.ac.jp

†These authors contributed equally to this work

Competing interest: The authors declare that no competing interests exist.

**Abstract** Photosynthesis is one of the most important reactions for sustaining our environment. Photosystem II (PSII) is the initial site of photosynthetic electron transfer by water oxidation. Light in excess, however, causes the simultaneous production of reactive oxygen species (ROS), leading to photo-oxidative damage in PSII. To maintain photosynthetic activity, the PSII reaction center protein D1, which is the primary target of unavoidable photo-oxidative damage, is efficiently degraded by FtsH protease. In PSII subunits, photo-oxidative modifications of several amino acids such as Trp have been indeed documented, whereas the linkage between such modifications and D1 degradation remains elusive. Here, we show that an oxidative post-translational modification of Trp residue at the N-terminal tail of D1 is correlated with D1 degradation by FtsH during high-light stress. We revealed that *Arabidopsis* mutant lacking FtsH2 had increased levels of oxidative Trp residues in D1, among which an N-terminal Trp-14 was distinctively localized in the stromal side. Further characterization of Trp-14 using chloroplast transformation in *Chlamydomonas* indicated that substitution of D1 Trp-14 to Phe, mimicking Trp oxidation enhanced FtsH-mediated D1 degradation under high light, although the substitution did not affect protein stability and PSII activity. Molecular dynamics simulation of PSII implies that both Trp-14 oxidation and Phe substitution cause fluctuation of D1 N-terminal tail. Furthermore, Trp-14 to Phe modification appeared to have an additive effect in the interaction between FtsH and PSII core in vivo. Together, our results suggest that the Trp oxidation at its N-terminus of D1 may be one of the key oxidations in the PSII repair, leading to processive degradation by FtsH.

## eLife assessment

This study adds a **fundamental** new perspective to a long-standing question: What controls the repair of photosystem II (PSII), a key process in maintaining and optimizing photosynthesis? The work supports a role for chemical modification in the recognition and subsequent degradation of a key protein subunit of PSII by a bacterial-type protease, suggesting that tryptophan oxidation of components of the photosynthetic apparatus after high light stress plays a critical role in initiating the PSII repair system. The evidence supporting the authors' conclusions is **solid**.

## Introduction

Light energy is essential for photosynthesis, which sustains our environment on Earth by generating oxygen and chemical energy for carbon fixation. The first step of photosynthetic electron transfer in the thylakoid membrane occurs at Photosystem II (PSII), where light energy is transferred to $P_{680}$ chlorophyll molecules to drive water oxidation, and electron is transferred to plastoquinone. PSII core complex is formed by two reaction center proteins D1 (PsbA) and D2 (PsbD), along with core antenna CP43 (PsbC) and CP47 (PsbB). However, light energy is known to cause photooxidative damage in PSII, especially reaction center protein D1. Photo-damaged D1 needs to be degraded to replace it with a newly synthesized one in the PSII repair (*Lindahl et al., 2000*; *Bailey et al., 2002*; *Silva et al., 2003*; *Kato et al., 2009*). When photooxidative damage in PSII exceeds the capacity of PSII repair, it ultimately leads to the status called 'photoinhibition' (*Aro et al., 1993*; *Murata et al., 2007*). PSII repair proceeds with the following steps, (i) oxidative damage to D1 protein in the PSII complex, (ii) migration of photo-damaged PSII laterally from grana stacks to non-appressed regions of thylakoid membranes, (iii) detachment of CP43 from photo-damaged PSII allows for access of protease degrading damaged D1, and (iv) concomitant D1 synthesis and reassembly of PSII into grana thylakoid. Substantial efforts to understand the mechanisms of PSII repair suggest that reversible phosphorylation of PSII core subunits is involved in fine-tuning the photo-damaged D1 turnover (*Barber et al., 2002*; *Fristedt et al., 2009*; *Kato and Sakamoto, 2014*).

Our previous studies along with those from other groups have shown that the fundamental D1 degradation in PSII repair is performed by FtsH, a membrane-bound ATP-dependent zinc metalloprotease that degrades membrane proteins in a processive manner, although several Deg proteases seem to facilitate the effective degradation by creating additional recognition sites for FtsH (*Lindahl et al., 2000*; *Bailey et al., 2002*; *Sakamoto et al., 2003*; *Silva et al., 2003*; *Kato et al., 2009*; *Kato and Sakamoto, 2014*). These proteins are universally conserved in prokaryotes and eukaryotic organelles (*Janska and Malgorzata Kwasniaka, 2013*). Photosynthetic organisms have hetero-hexameric FtsH complex in the thylakoid membrane, which is composed of type A and type B subunits (*Kato and Sakamoto, 2018*; *Yi et al., 2022*). In the thylakoid membrane of *Arabidopsis thaliana*, for example, FtsH1 or FtsH5 (type A) and FtsH2 or FtsH8 (type B) form functional FtsH complex (*Yu et al., 2004*; *Yu et al., 2005*; *Zaltsman et al., 2005*). *Arabidopsis* mutants lacking FtsH5 or FtsH2 are, though viable, highly vulnerable to PSII photodamage under high light and are named *yellow variegated1* (*var1*) and *var2* from the characteristic variegated phenotype (*Chen et al., 2000*; *Takechi et al., 2000*; *Sakamoto et al., 2002*). This variegated phenotype implies that FtsH is required for proper protein quality control during proplastid-to-chloroplast differentiation in seed plants (*Miura et al., 2007*; *Sakamoto et al., 2009*). Similarly, the FtsH complex in the thylakoid membranes of *Chlamydomonas reinhardtii* consists of FtsH1 (type A) and FtsH2 (type B) (*Malnoë et al., 2014*). *Chlamydomonas* studies revealed that FtsH is also involved in the degradation of cytochrome $b_6f$ complex and light-harvesting antenna of photosystem I (PSI) (*Malnoë et al., 2014*; *Bujaldon et al., 2017*). Furthermore, recent studies suggest that increased turnover of FtsH is crucial for their function under high-light stress. That is compensated by upregulated *FtsH* gene expression (*Wang et al., 2017*; *Kato et al., 2018*).

In *Escherichia coli*, FtsH was shown to recognize either N- or C-terminal tail of membrane protein substrates to pull their substrates and push them into the protease chamber by ATPase activity (*Ito and Akiyama, 2005*). Similarly, the fact that both D1's N-terminal end and the catalytic site of FtsH are exposed to the stromal side implies D1 to be recognized by FtsH from its N-terminus (*Umena et al., 2011*). Supporting this, processive degradation of D1 by FtsH was shown to be attenuated by the loss of N-terminal tail of D1 (*Komenda et al., 2007*; *Michoux et al., 2016*). While these observations

suggest involvement of the N-terminal region in D1, whether D1 undergoes oxidative modification at its N-terminal region has not been investigated.

What is the disorder of photosystems leading to photo-oxidative damage in PSII? Light energy frequently leads to the generation of reactive oxygen species (ROS) such as singlet oxygen at around PSII (*Ohnishi et al., 2005*; *Tyystjarvi, 2008*; *Yamamoto et al., 2008*), which may cause oxidative post-translational modification (OPTM) of subunit proteins (*Li and Kim, 2022*). It is noteworthy that light-dependent oxidation of amino acids, either in free forms or as peptide residues, has been reported, including thiol-containing (Cys and Met) and aromatic (Tyr, Phe, Trp) amino acids. For example, Cys and Met are prone to oxidation, whereas the oxidized Cys and Met can be reduced enzymatically. In contrast to these reversible OPTMs, OPTM of Trp is irreversible (*Rinalducci et al., 2008*; *Ehrenshaft et al., 2015*). Thus, the only way to remove irreversible oxidized residues is proteolytic degradation, implying that Trp oxidation might trigger D1 degradation, either directly or indirectly in the PSII repair. As summarized in *Figure 1A*, oxidation of Trp side chain results in the formation of oxindolylalanine (OIA), N-formylkynurenine (NFK), and kynurenine (KYN). ROS attacks and opens the pyrrole ring of Trp, and forms a di-oxidized Trp derivative, NFK. Indeed, Trp residues in photosynthetic protein components were shown to be oxidized both in vitro and in vivo (*Anderson et al., 2002*; *Dreaden et al., 2011*; *Dreaden Kasson et al., 2012*). However, although oxidative modification of D1 and other subunits has been documented previously, how these molecules are recognized and undergo degradation remains elusive. In this study, we investigated whether Trp oxidation in PSII core proteins influences D1 degradation mediated by FtsH. Our integrative approaches to address this question, by mass-spectrometry, site-directed mutagenesis, D1 degradation assay, and simulation model suggest that an N-terminal Trp oxidation is likely to be a key OPTM to trigger D1 degradation in the PSII repair.

## Results

### Increased OPTM of Trp residues in *Arabidopsis var2* mutant

Previous studies using isolated spinach thylakoid membranes and *Arabidopsis* seedlings revealed several Trp residues oxidized in PSII core proteins, as summarized in *Figure 1B* and *Table 1* (*Rinalducci et al., 2008*; *Dreaden et al., 2011*; *Dreaden Kasson et al., 2012*). Trp-oxidized derivatives, OIA, NFK, and KYN, give the shifts of peptide mass to +16,+32, and +4 Da, respectively (*Figure 1A*). In this study, we attempted to assess if the oxidation of certain Trp residues is associated with D1 degradation in the PSII repair cycle. To investigate this, comprehensive detection of Trp oxidation within protein extracts in *Arabidopsis* has been established using label-free quantitative mass-spectrometry, as previously reported (*Dogra et al., 2019*). First, we characterized Trp oxidation from total proteins of *Arabidopsis* wild-type seedlings grown in continuous light (100 µmol photons $m^{-2}s^{-1}$). Consistent with previous results (*Dreaden Kasson et al., 2012*; *Dogra et al., 2019*), two Trp residues in D1, namely Trp-14 and Trp-317, were shown to be oxidized (*Figure 1—figure supplement 1* and *Figure 1—figure supplement 2*). The total sequence coverage obtained by our mass spectrometry for D1 was 26%. Further mass spectrometry in extracts of *Arabidopsis* mutants *var2* lacking FtsH2, which is shown to impair D1 degradation and exhibit substantial accumulation of ROS (*Kato et al., 2009*), revealed the accumulation of oxidized Trp in the PSII complex. The levels of Trp-oxidized derivatives, OIA, NFK, and KYN in Trp-14 and the level of KYN in Trp-314 were significantly increased in *var2* compared to the wild type, respectively (*Figure 1C*). These results prompted us to characterize the role of these Trp oxidations in the PSII repair further.

### OPTM of Trp residues in *Chlamydomonas* PSII core proteins

To validate whether Trp oxidation detected in *Arabidopsis* seedings is also detectable in other model organisms, we investigated extracts from *Chlamydomonas* thylakoid membranes (*Table 2*). Mass spectrometry following trypsin digestion demonstrated that several Trp residues in PSII were oxidized; among the core subunits, D1 had oxidation of four residues, D2 had three, CP43 had five, and CP47 had three (*Table 2* and *Figure 1—figure supplement 3*). The position of oxidized Trp residues in their amino acid sequence was shown in *Figure 1B* for D1 and *Figure 1—figure supplement 3* for D2, CP43, and CP47. Together with previous studies, Trp-14 and Trp-317 in D1, Trp-21 and Trp-328 in D2, Trp-353 and Trp-375 in CP43, and Trp-275 and Trp-302 in CP47 were commonly identified among at least two organisms. Spatial arrangement of these Trp residues that were commonly oxidized in

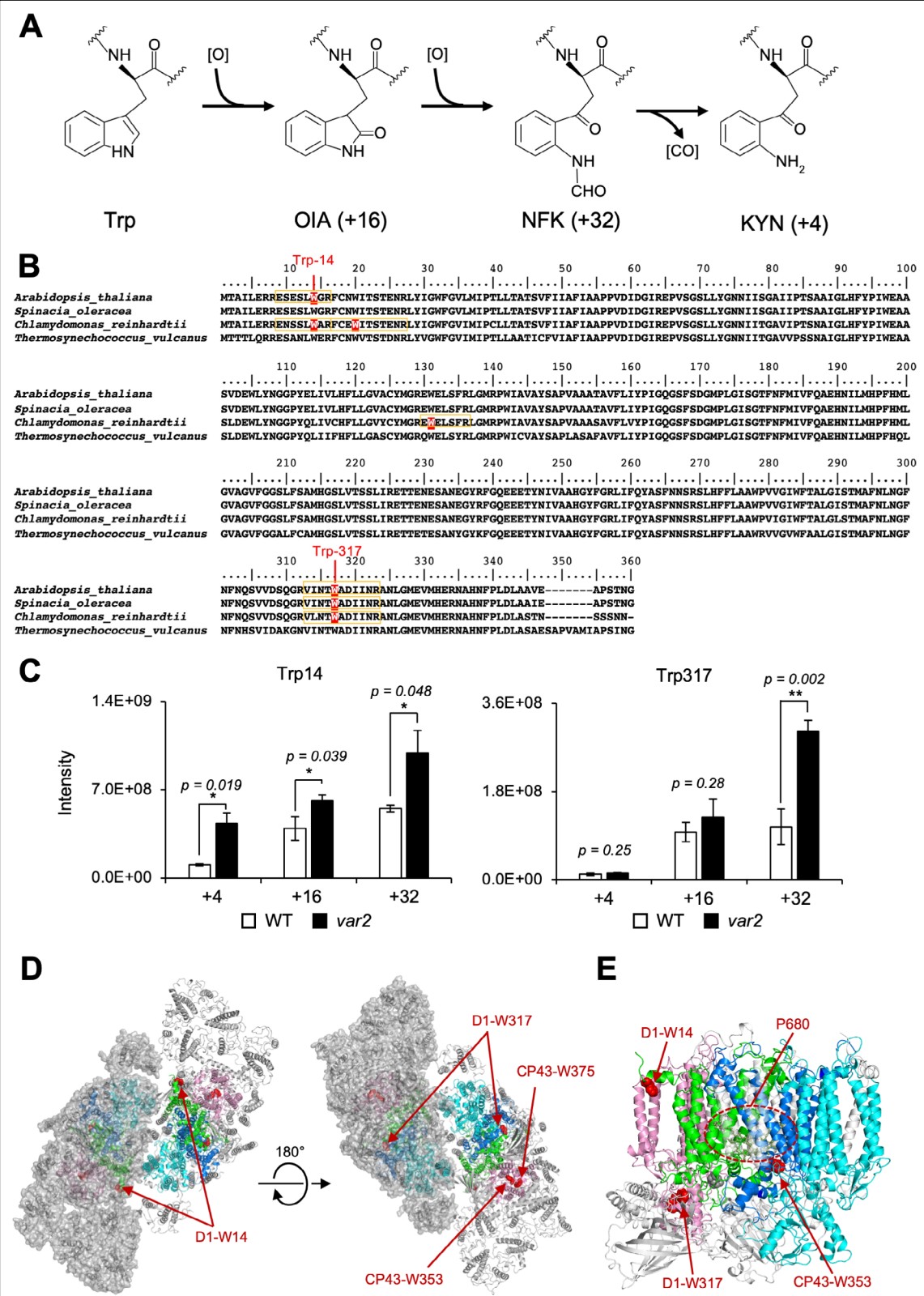

**Figure 1.** The Oxidized Trp residues in PSII complex. (**A**) Trp-oxidation pathway. OIA: oxindolylalanine, NFK: N-formylkynurenine, KYN: Kynurenine. (**B**) Multiple alignment of D1 protein from *Arabidopsis*, spinach, *Chlamydomonas*, and *Thermosynechococcus vulcanus*, showing oxidized Trp residues. Orange color boxes indicate the identified peptide by the MS-MS analysis. Oxidized Trp residues are highlighted in red. (**C**) Oxidation levels of three oxidative variants of Trp in Trp14 and Trp317 containing peptides in *var2* and WT obtained by label-free MS analysis. The abundance of oxidized variants

*Figure 1 continued on next page*

*Figure 1 continued*

(+4: KYN,+16: OIA, and +32: NFK) of Trp14 and Trp317 were calculated using the intensity values. Asterisks indicate statistically significant differences between the mean values (*<0.05, **<0.01; Student's t-test). (**D–E**) Structural positions of oxidized Trp residues in PSII core proteins. The side chain of oxidized Trp residues are shown with red-colored space-filling model and indicated with arrows. The P680 special chlorophyll pair is indicated with dark-green colored ball-stick model in panel e. PSII dimer (panel d) and monomer (panel e) from *Chlamydomonas reinhardtii* (PDB ID is 6KAC) is shown in cartoon model without cofactors Top view from stromal side or lumenal side (**D**) and the side view from the dimer interface (**E**) are shown respectively. The color code of each subunit is, Green, D1; Dark blue, D2; Purple, CP43, Cyan, CP47. Protein structure graphics were generated with PyMOL ver. 2.4.0 software.

The online version of this article includes the following video, source data, and figure supplement(s) for figure 1:

**Source data 1.** Trp oxidation in *Chlamydomonas* PSII core proteins.

**Figure supplement 1.** Detected oxidative modifications at Trp14 of D1 in *Arabidopsis thaliana*.

**Figure supplement 2.** Detected oxidative modifications at Trp317 of D1 in *Arabidopsis thaliana*.

**Figure supplement 3.** Positions of oxidized Trp residues in the identified peptide of PSII core complex by the MS-MS analysis.

**Figure supplement 4.** Structural positions of oxidized Trp residues in PSII core proteins.

**Figure 1—animation 1.** Oxidized Trp residues assigned in the PSII dimer.

*Arabidopsis* and *Chlamydomonas* was compared within the structure of PSII complex, as shown in *Figure 1D and E* (oxidized Trp residues were assigned in the PSII dimer, and its 3D image is shown in *Figure 1—animation 1*). Also, oxidized Trp residues were assigned in the PSII structure from *Thermosynecoccocus vulcanus* and were shown in *Figure 1—figure supplement 4*. Intriguingly, most of these oxidized Trp residues are positioned at the lumenal side of the PSII core complex, which appeared to surround the $Mn_4O_5Ca$ cluster in the PSII structure model. In contrast, two oxidized Trp residues, Trp-14 in D1 and Trp-21 in D2 close to the N-terminus of the polypeptides, are located on the stromal side. The fact that the oxidized Trp residues are predominantly observed around the $Mn_4O_5Ca$ cluster may reflect photoinhibition of PSII electron donor side and concomitant ROS generation. In contrast, stromal Trp oxidation is novel and localized at the N-terminal alpha-helix, which might suggest its effects in processive D1 degradation.

## Site-directed mutagenesis of Trp residues undergoing OPTM in D1

To test whether any change in the oxidized Trp residues is associated with D1 degradation, we performed site-directed mutagenesis using chloroplast transformation in *Chlamydomonas*, to substitute the corresponding Trp for other amino acids in D1. Based on the mass-spectrometric results, we focused on Trp-14 and Trp-317, each of which was replaced by Ala (non-polar and hydrophobic) or Phe (aromatic and hydrophobic), respectively. The vectors harboring spectinomycin/streptomycin-resistant *aadA* cassette and the mutated *psbA* gene were transformed into Δ*psbA* mutant Fud7 (*Figure 2A*). Transformants were selected on mixotrophic Tris-acetate-phosphate (TAP) plates containing spectinomycin, and their homoplasmicity was subsequently confirmed by PCR using specific primers and sequencing. All transformants grew like the control strain on mixotrophic TAP plates (*Figure 2B*). However, the transformants in which Trp-14 or Trp-317 was substituted to Ala, (W14A, W317A, and W14A/W317A) showed significantly impaired growth on photoautotrophic high salt minimal (HSM) plates. Ala substitution at both Trp-14 and Trp-317 led to decreased photosynthetic activities due to reduced accumulation of D1 and other PSII core proteins (*Figure 2B, C and D*), indicating its defect in stability and/or the translation of D1 protein.

   In contrast, Phe substitution at the same sites had little effect on their growth under growth light (30 μmol photons $m^{-2}s^{-1}$). These transformants (W14F, W317F, and W14F/W317F) accumulated PSII core proteins whose amounts were comparable to the control levels. They did not show a substantial change in photosynthetic activities as evidenced by comparable electron transport rates through the PSII complex and oxygen-evolving activity (*Figure 2C and E*). We next examined their photoautotrophic growth under high light (320 μmol photons $m^{-2}s^{-1}$). Under this condition, however, W14F exhibited significantly impaired growth, and W317F grew slightly slower than control cells. Double mutant W14F/W317F synergistically increased high-light sensitivity but the growth defect appeared to be similar to W14F, suggesting that Phe substitution at Trp-14, but not at Trp-317 had profound effects in the PSII repair cycle (*Figure 2B* and *Figure 2—figure supplement 1*).

**Table 1.** Modification reported in Trp in PSII core proteins.

| Organism | Protein | Sequence | Modified Trp residue | Oxidation status | Position in Chlamydomonas | Reference |
|---|---|---|---|---|---|---|
| | D1 | VINT(W*)ADIINR | Trp317 | OIA, NFK, KYN | Trp317 | |
| | D2 | FTKDEKDLFDSMDD(W*)LR | Trp22 | OIA, NFK, KYN | Trp21 | |
| | | DLFDSMDD(W*)LR | Trp22 | OIA, KYN | Trp21 | |
| | CP43 | AP(W*)LEPLR | Trp365 | OIA, NFK, KYN | Trp353 | *Dreaden Kasson et al., 2012* |
| | | AP(W*)LEPLRGPNGLDLSR | Trp365 | OIA, NFK, KYN | Trp353 | |
| | | AP(W*)LEPLR | Trp365 | OIA, KYN | Trp353 | *Anderson et al., 2002* |
| | | F(W*)DLR | Trp359 | OIA | Trp347 | *Dreaden Kasson et al., 2012* |
| Spinach | | DIQP(W*)QER | Trp387 | OIA | Trp375 | |
| | D1 | ESESL(W*)GR | Trp14 | OIA, NFK, KYN | Trp14 | |
| | | VINT(W*)ADIINR | Trp317 | OIA, NFK, KYN | Trp317 | |
| | D2 | DLFDIMDD(W*)LR | Trp22 | OIA, NFK, KYN | Trp21 | |
| | | A(W*)MAAQDQPHENLIFPEEVLPR | Trp329 | OIA, NFK, KYN | Trp328 | |
| | CP43 | AP(W*)LEPLR | Trp365 | OIA, NFK, KYN | Trp353 | |
| | | DIQP(W*)QER | Trp387 | OIA, NFK, KYN | Trp375 | |
| | CP47 | YQ(W*)DQGYFQQEIYR | Trp275 | OIA, NFK, KYN | Trp275 | |
| *Arabidopsis* | | VSAGLAENQSLSEA(W*)AK | Trp302 | OIA, NFK, KYN | Trp302 | *Dogra et al., 2019* |

OIA: oxindolylalanine, NFK: N-formylkynurenine
KYN: Kynurenine

## Site-directed mutagenesis of Trp residues in CP43

We next examined high-light sensitivity in the CP43 Trp mutants. As an important step in the PSII repair, PSII complex is partially disassembled by CP43 detachment, and this process likely allows FtsH to access photo-damaged D1. Therefore, Trp oxidation in CP43 may play a role in PSII disassembly and D1 degradation concomitantly. To test this, we substituted Trp-353 and Trp-375 for either Ala or Phe as carried out in D1. Transformants were generated by cotransformation of Fud7, using the vector harboring wild-type *psbA* gene and the vector harboring the mutated *psbC* gene. Consequently, we obtained four single mutants (W353A, W353F, W375A, and W375F) and two double mutants (W353A/W375A and W353A/W375F; *Figure 3*). Mixotrophic growth on TAP plates was comparable among all transformants and control cells (*Figure 3*). All the transformants except for W353A/W375A grew normally on the phototrophic condition under growth light condition. Supporting normal growth, immunoblot analysis showed normal accumulation of PSII core proteins, D1 and CP43, in all lines except for the double mutant W353A/W375A. On the other hand, D1 and CP43 were severely reduced in W353A/W375A, indicating that Ala substitution in these residues resulted in highly unstable or impaired PSII complex formation (*Figure 3*). Further analysis of these mutants under high light showed

**Table 2.** Trp oxidation in *Chlamydomonas* PSII core proteins.

| Accession | Protein | Sequence | range | Modified Trp residue | Oxidation status |
|---|---|---|---|---|---|
| DAA00922.1_20 | D1 | ENSSL(W*)AR | 9–16 | Trp14 | OIA, NFK, KYN |
| | | FC$^{cam}$E(W*)ITSTENR | 17–27 | Trp20 | OIA, NFK, KYN |
| | | E(W*)WELSFR | 130–136 | Trp131 | OIA, NFK, KYN |
| | | VLNT(W*)ADIINR | 313–323 | Trp317 | OIA, NFK, KYN |
| DAA00964.1_63 | D2 | T(W*)FDDADDWLR | 13–23 | Trp14 | OIA, NFK, KYN |
| | | TWFDDADD(W*)LR | 13–23 | Trp21 | OIA, NFK, KYN |
| | | T(W*)FDDADD(W*)LR | 13–23 | Trp14, Trp21 | OIA, NFK, KYN |
| | | A(W*)MAAQDQPHER | 327–338 | Trp328 | OIA, NFK, KYN |
| | | A(W*)M$^{ox}$AAQDQPHER | 327–338 | Trp328 | OIA, NFK, KYN |
| DAA00966.1_65 | CP43 | DQETTGFA(W*)WSGNAR | 15–29 | Trp23 | OIA, NFK, KYN |
| | | DQETTGFAW(W*)SGNAR | 15–29 | Trp24 | OIA, NFK, KYN |
| | | DQETTGFA(W*)(W*)SGNAR | 15–29 | Trp23, Trp24 | OIA, NFK, KYN |
| | | AM$^{ox}$YFGGVYDT(W*)APGGGDVR | 167–185 | Trp177 | OIA, NFK, KYN |
| | | GP(W*)LEPLR | 351–358 | Trp353 | OIA, NFK, KYN |
| | | NDIQP(W*)QER | 370–378 | Trp375 | OIA, NFK, KYN |
| DAA00933.1_31 | CP47 | YQ(W*)DQGFFQQEIQK | 273–286 | Trp275 | OIA, NFK, KYN |
| | | VQASLAEGASLSDA(W*)SR | 288–304 | Trp302 | OIA, NFK, KYN |
| | | TGAM$^{ox}$NSGDGIAVG(W*)LGHASFK | 327–347 | Trp340 | OIA, NFK, KYN |

C$^{cam}$ = Cys carbamidomethylation. W* = Trp oxidative modifications. M$^{ox}$ = Met oxidation.

that the transformants except for W353F did not increase high-light sensitivity on their growth. Only W353F, but not W353F/W375F, showed a slight reduction in their growth under high-light conditions. This is consistent with a previous study (*Anderson et al., 2002*). However, the impaired growth of W353F was not as severe as that of D1 W14F or W14F/W317F, suggesting its effect was limited. At least in our site-directed mutagenesis, Trp oxidation in CP43 appeared to have little impact on the PSII repair.

## Substitution of Trp-14 with Phe accelerates D1 degradation

To evaluate whether Trp substitution in D1 affects PSII damage or repair, we next measured the maximum quantum yield of PSII ($F_V/F_M$) and subsequently monitored D1 levels under growth or high-light conditions. Trp-substituted lines grown in TAP medium under growth light were pre-incubated

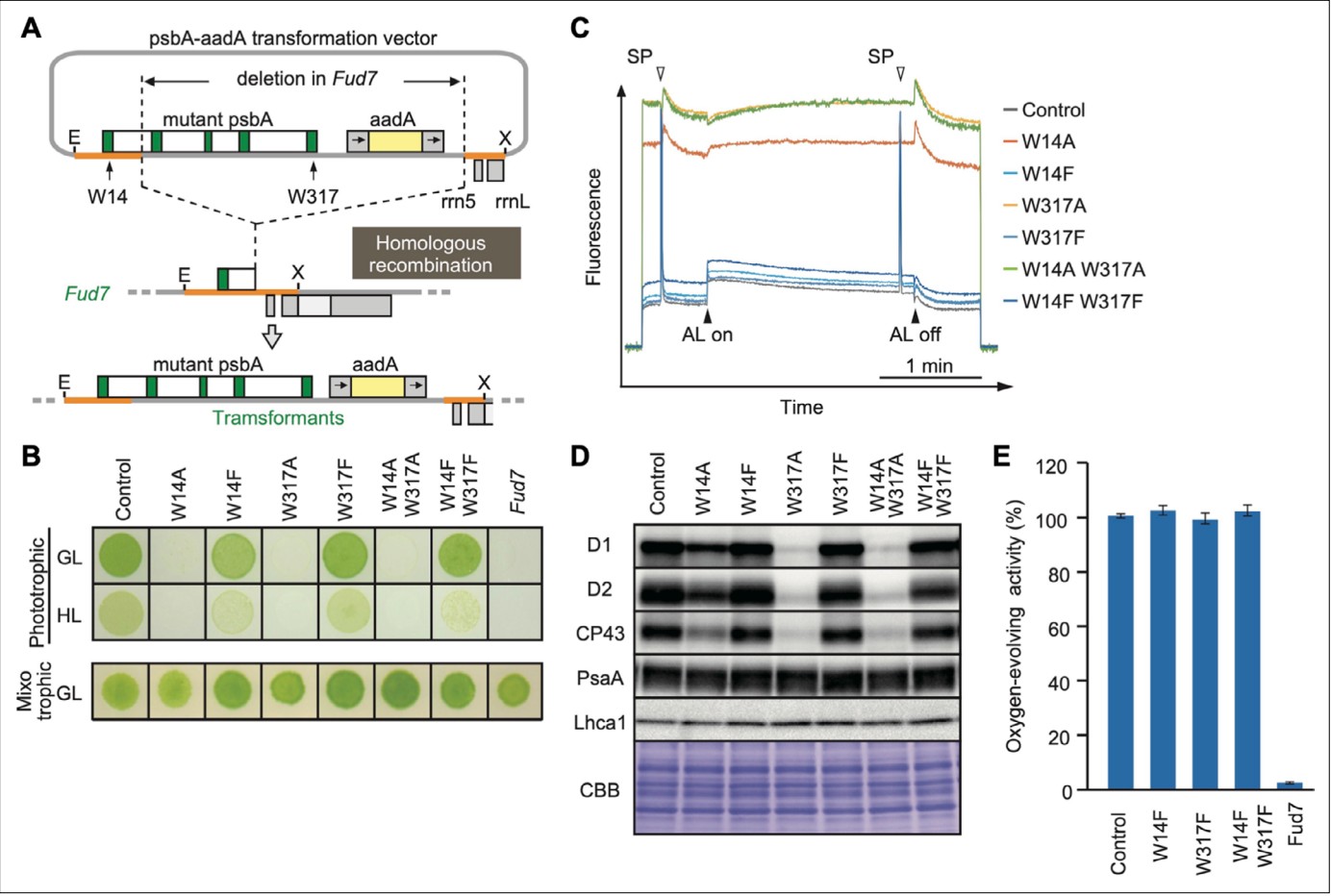

**Figure 2.** High-light sensitive phenotype in the *Chlamydomonas* D1 transformants in which Trp-14 and Trp317 were mutated. (**A**) Schematic drawing of the transforming vector carrying *psbA*, its flanking regions of the chloroplast DNA, and the selectable *aadA* marker cassette. E and X represent restriction sites of *EcoR*I and *Xho*I, respectively. Green boxes represent exons 1–5 of *psbA*. Fud7 is the *psbA* deletion mutant of *Chlamydomonas*. (**B**) Phototrophic growth of Trp-substituted transformants on HSM medium and mixotrophic growth on TAP medium. GL, growth light (30 μmol photons m$^{-2}$s$^{-1}$): HL, high light (320 μmol photons m$^{-2}$s$^{-1}$). (**C**) Chlorophyll fluorescence induction kinetics in Trp-substituted transformants. SP, saturating pulse. AL, actinic light. (**D**) Protein accumulation in the transformants. Thylakoid proteins of cells grown in TAP medium under growth-light condition were separated by SDS-PAGE and analyzed by immunoblotting with antibodies against PSII subunits (D1, D2, and CP43), PSI subunits (PsaA), and light-harvesting complex of PSI (Lhca1). (**E**) Oxygen-evolving activity of the transformants.

The online version of this article includes the following source data and figure supplement(s) for figure 2:

**Source data 1.** Source data for western blot images in *Figure 2*.

**Figure supplement 1.** High-light sensitive phenotype in the *Chlamydomonas* D1 transformants in which Trp-14 and Trp317 were mutated.

in the presence or absence of chloramphenicol (CAM), an inhibitor of chloroplast protein synthesis. CAM blocks the PSII repair at the step of D1 synthesis and allows us to evaluate photodamage and D1 degradation. Cells incubated under growth light or high light were subjected to chlorophyll fluorescence measurement and immunoblot analysis. Under growth light condition and in the absence of CAM, both PSII activity ($F_V/F_M$ values) and D1 levels were comparable among all Trp-substituted lines and the control (*Figure 4A*). This result was consistent with their photoautotrophic growth under growth light (*Figure 2B*). When CAM was added, D1 levels decreased only slightly during incubation (90 min) in the control. D1 degradation rate was comparable in all Trp-substituted lines and control (*Figure 4C*), indicating that all Trp-substituted D1 proteins formed stable and functional PSII complex under growth light.

Under high-light condition, however, $F_V/F_M$ values in Trp-substituted lines significantly decreased, compared to that observed in the control even in the absence of CAM (*Figure 4B*). These vulnerabilities to high light were consistent with their impaired growth under high light (*Figure 2B*).

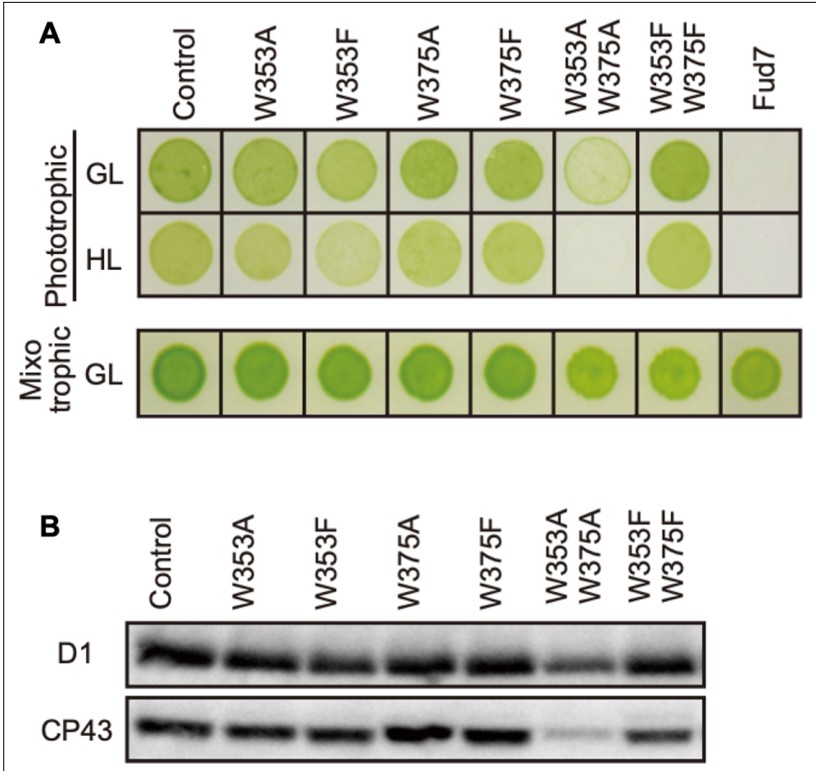

**Figure 3.** Characterization of Chlamydomonas CP43 transformants in which Trp-353 and Trp-375 were mutated. (**A**) Phototrophic growth on HSM medium and mixotrophic growth on TAP medium at growth light (GL) at 30 μmol m$^{-2}$s$^{-1}$ or high light, (HL) at 320 μmol m$^{-2}$s$^{-1}$. (**B**) Protein accumulation in the transformants. Thylakoid proteins of cells grown in TAP medium under growth light condition were separated by SDS-PAGE and analyzed by immunoblotting with antibodies against PSII subunits (**D1 and CP43**).

The online version of this article includes the following source data for figure 3:

**Source data 1.** Source data for western blot images in *Figure 3*.

To our surprise, D1 levels in W14F and W14F/W317F concomitantly decreased during high-light incubation (*Figure 4B*). In contrast, D1 levels in W317F were similar to those in control cells. When the PSII repair engages properly, high-light irradiation does not alter D1 levels because rapid D1 synthesis compensates turnover of photo-damaged D1. Given decreased D1 under high light, W14F was likely to cause faster D1 degradation. To confirm this possibility, D1 degradation in the presence of CAM was measured. PSII activity in all Trp-substituted lines fell at similar rates compared with control cells in the presence of the CAM (*Figure 4D*), indicating the light-induced damage was at the similar level among all Trp-substituted lines and the control. In contrast, our time course experiment indicated that W14F and W14F/W317F decreased D1 faster than the control and W317F (*Figure 4D*); the D1 level in W14F and W14F/W317F decreased approximately 60% and 50% of the initial level, respectively, whereas those in control cells and W317F remained 80% (*Figure 4D*).

Based on these D1 degradation assays, we assumed that D1 degradation by proteolysis was enhanced by W14F substitution, despite the fact that PSII suffered from photodamage equally among other lines and the control during high-light irradiation. To exclude the possibility that W14F decelerates D1 synthesis rather than accelerating degradation, we analyzed protein synthesis in Trp-substituted lines by in vivo pulse labeling in the presence of cycloheximide, which prevents the synthesis of the nuclear-encoded proteins. As shown in *Figure 5*, D1 synthesis was shown to proceed comparably in all lines. Collectively, our findings demonstrated that Trp-14 substitution to Phe enhanced D1 degradation, whereas it affected neither the light-induced damage in PSII, D1 synthesis, nor PSII stability.

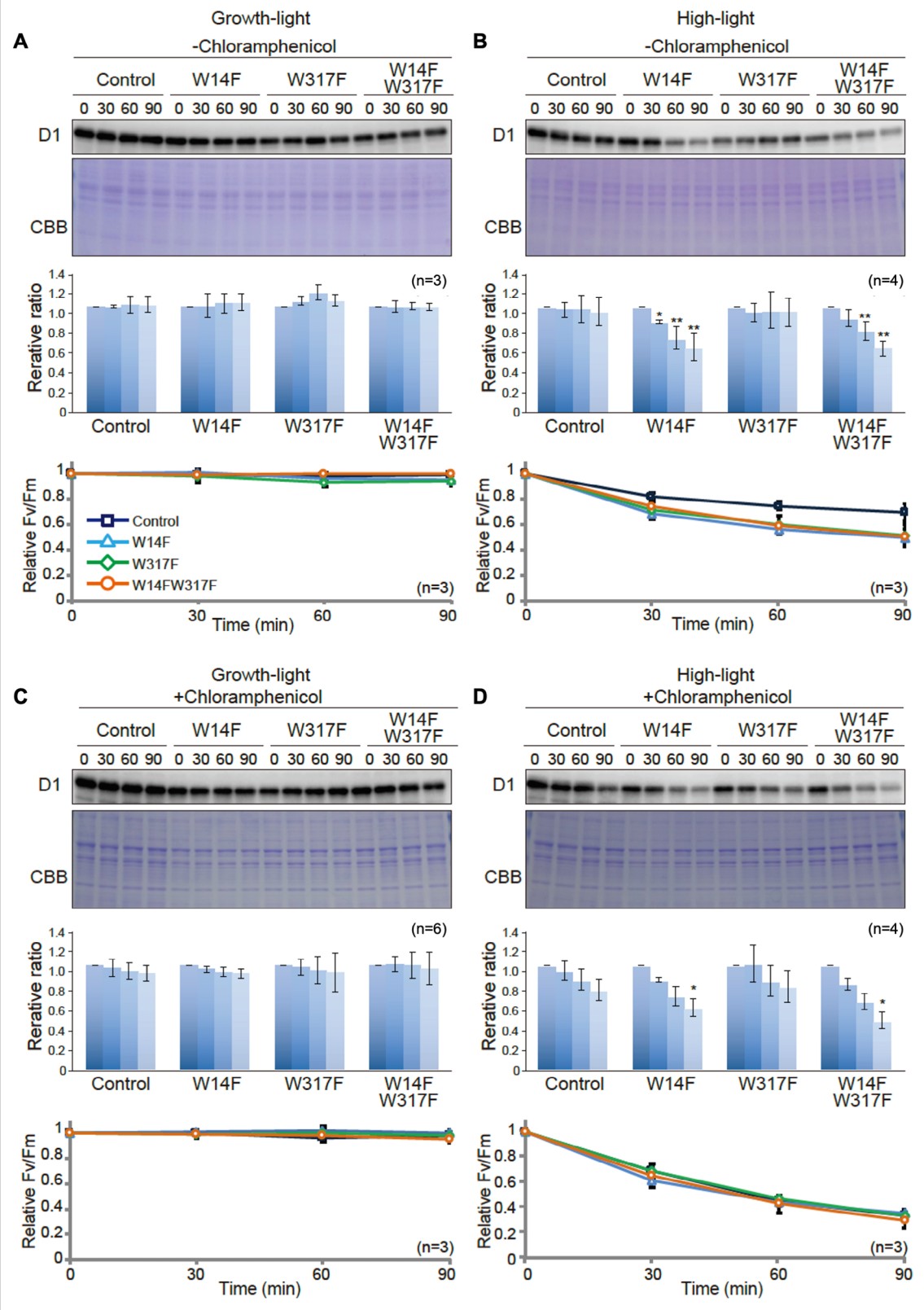

**Figure 4.** D1 degradation assay in W14F and W317F transformants demonstrating enhanced D1 degradation under high-light stress. The transformants were incubated under high-light (320 μmol photons m$^{-2}$s$^{-1}$) or growth-light (30 μmol photons m$^{-2}$s$^{-1}$) conditions in the absence or presence of inhibitor of chloroplast protein synthesis, CAM, and subjected to D1 degradation assay. (**A**) Growth-light in the absence of CAM; (**B**) high-light in the absence of CAM; (**C**) growth-light in the presence of CAM; (**D**) high-light in the presence of CAM. Immunoblot results of D1 in the transformants are shown at the

*Figure 4 continued on next page*

*Figure 4 continued*

top of each panel. A representative immunoblot using anti-D1 is depicted. Quantified D1 levels using NIH Image program are shown in the middle. Values are means ± SD. Asterisks indicate statistically significant differences between the mean values (*<0.05, **<0.01; Student's t-test). Time course analysis of maximal photochemical efficiency of PSII, $F_V/F_M$, are shown at the bottom.

The online version of this article includes the following source data for figure 4:

**Source data 1.** Source data for western blot images and graphs in *Figure 4*.

## Enhanced D1 degradation due to the substitution of Trp-14 is mitigated in the *ftsH* mutant

To address whether the increased D1 degradation in W14F (and W14F/W317F) involved proteolysis by FtsH, these Trp substitutions were introduced into an *ftsH* mutant deficient in thylakoid FtsH activity. In the thylakoid membrane of *Chlamydomonas*, a hetero-oligomeric FtsH complex composed of FtsH1 (type-A) and FtsH2 (type-B) exists, and the *ftsh1-1* mutant, expressing an inactive FtsH1 due to the amino-acid substitution in the ATP-binding domain, has been reported (*Malnoë et al., 2014*). We performed mating W14F and W14F/W317F transformants (mt+) with *ftsh1-1* (mt-), and the resulting mutants, W14F *ftsh1* and W14F/W317F *ftsh1*, were subjected to D1 degradation assay. The results indicated that as expected, the enhanced D1 degradation observed in W14F and W14F/W317F cells was partially mitigated under *ftsh1-1* background (*Figure 6*), when CAM was added. These results suggested that FtsH plays a key role in the increased D1 degradation in W14F and W14F/W317F.

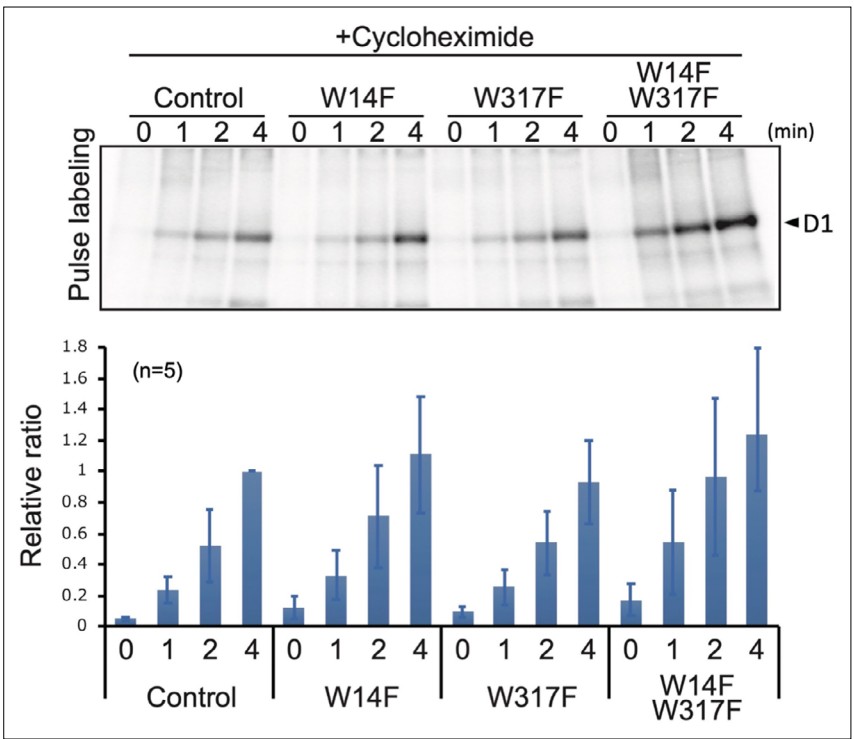

**Figure 5.** Protein synthesis in the transformants studied by in vivo protein labeling. Cells were radio-labeled in vivo with [35]S, in the presence of cycloheximide for 1, 2, and 4 min. Total proteins were separated by SDS-PAGE. The bands corresponding to D1 is indicated by arrowheads. Quantified newly synthesized D1 levels using the Image J program are shown in bottom panels. To normalize values from four independent experiments, the ratio of control at 4 min was adjusted as 1, and the relative ratios are indicated. Values are means ± SD.

The online version of this article includes the following source data for figure 5:

**Source data 1.** Source data for the pulse-labeling image and the graph in *Figure 5*.

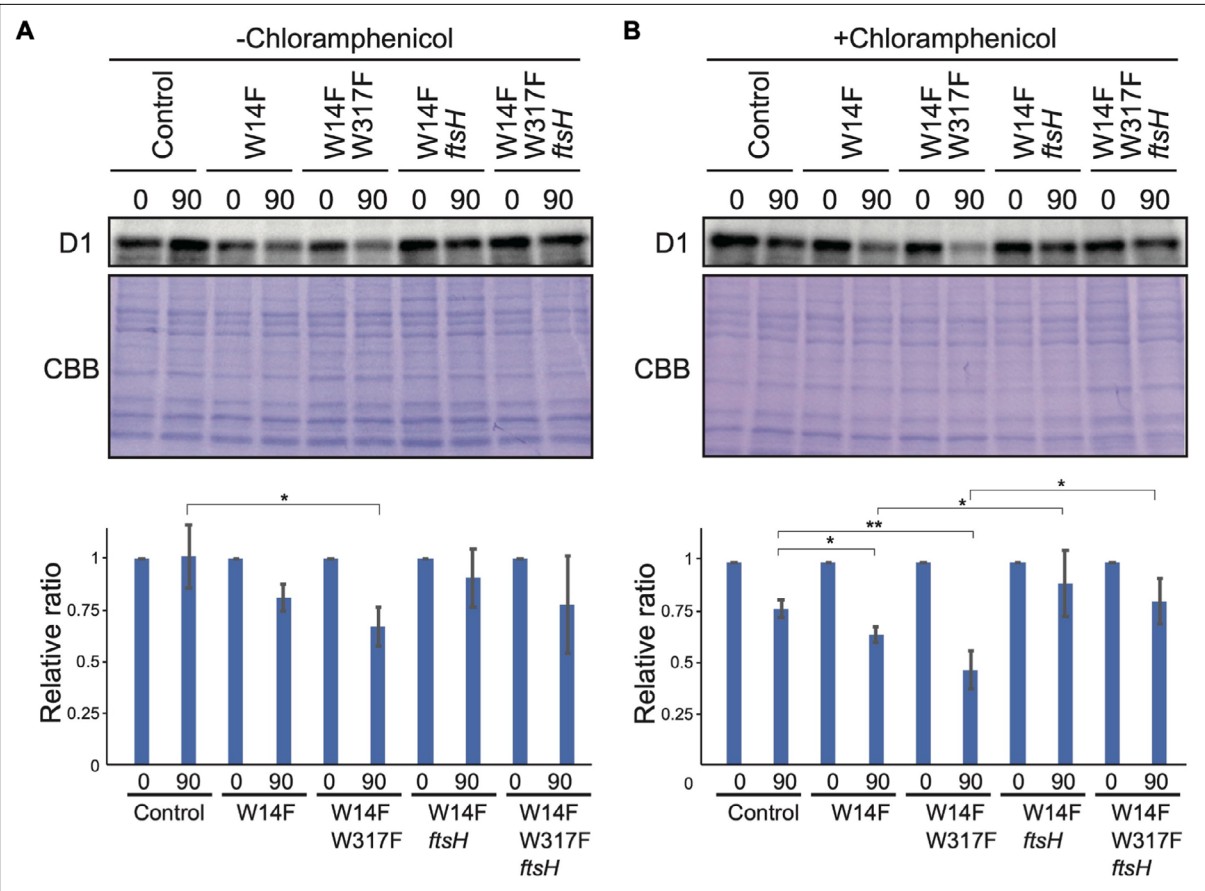

**Figure 6.** D1 degradation assay in W14F and W14F/W317F transformants in the *ftsh* mutant background. Rate of D1 degradation in the W14F *ftsH* and W14F/W317F *ftsH* was investigated as shown in *Figure 3*. Cultured cells were incubated under high-light conditions (320 µmol photons m⁻²s⁻¹) in the absence (**A**) or presence (**B**) of CAM. Signals of immunoblots were quantified using NIH Image program. Values are means ± SD (n=4). A representative immunoblot using anti-D1 is depicted. Values are means ± SD. Asterisks indicate statistically significant differences between the mean values (*<0.05, **<0.01; Student's t-test).

The online version of this article includes the following source data for figure 6:

**Source data 1.** Source data for western blot images and graphs in *Figure 6*.

## Molecular dynamics simulation suggests W14F mimicking Trp-14 oxidation

Although our site-directed mutagenesis in Trp-14 showed its effect in D1 degradation, how Trp oxidation can be structurally correlated with Trp to Phe mutagenesis should be taken into consideration. To investigate this, we employed molecular dynamics (MD) simulation, a powerful tool to simulate movements of amino acids in a protein complex, using the crystal structure of PSII complex from *Thermosynechococcus vulcanus* (*Sakashita et al., 2017b*; *Sakashita et al., 2017a*; *Kawashima et al., 2018*). D1 Trp-14 is located in the first α-helix at the N-terminus and hydrogen-bonded with PsbI Ser-25 (*Figure 7A*). It is deduced that this hydrogen bond restricts the conformational change around D1 Trp-14 and limit the fluctuation of D1 N-terminus. The simulation indicated that the hydrogen bond disappeared (*Figure 7B and C*) and the structural fluctuation of D1 Trp-14 was increased as compared with WT (*Figure 7D*) when Trp-14 is oxidized to NFK or is replaced with Phe residue. The increased fluctuation of the side chain also influences the Cβ-Cβ distance between D1 Trp-14 and PsbI Ser-25; the two Cβ atoms became farther away from each other when D1 Trp-14 is oxidized to NFK (*Figure 7E*). Of note, the amino acid substitution on Trp-14 to Phe showed similar trends as those observed when D1-Trp-14 is oxidized to NFK. These results suggest that the structural change of Trp-14 affect the local movement. The increased fluctuation of the first α-helix of D1 would give a chance to recognize the photo-damaged D1 by FtsH protease.

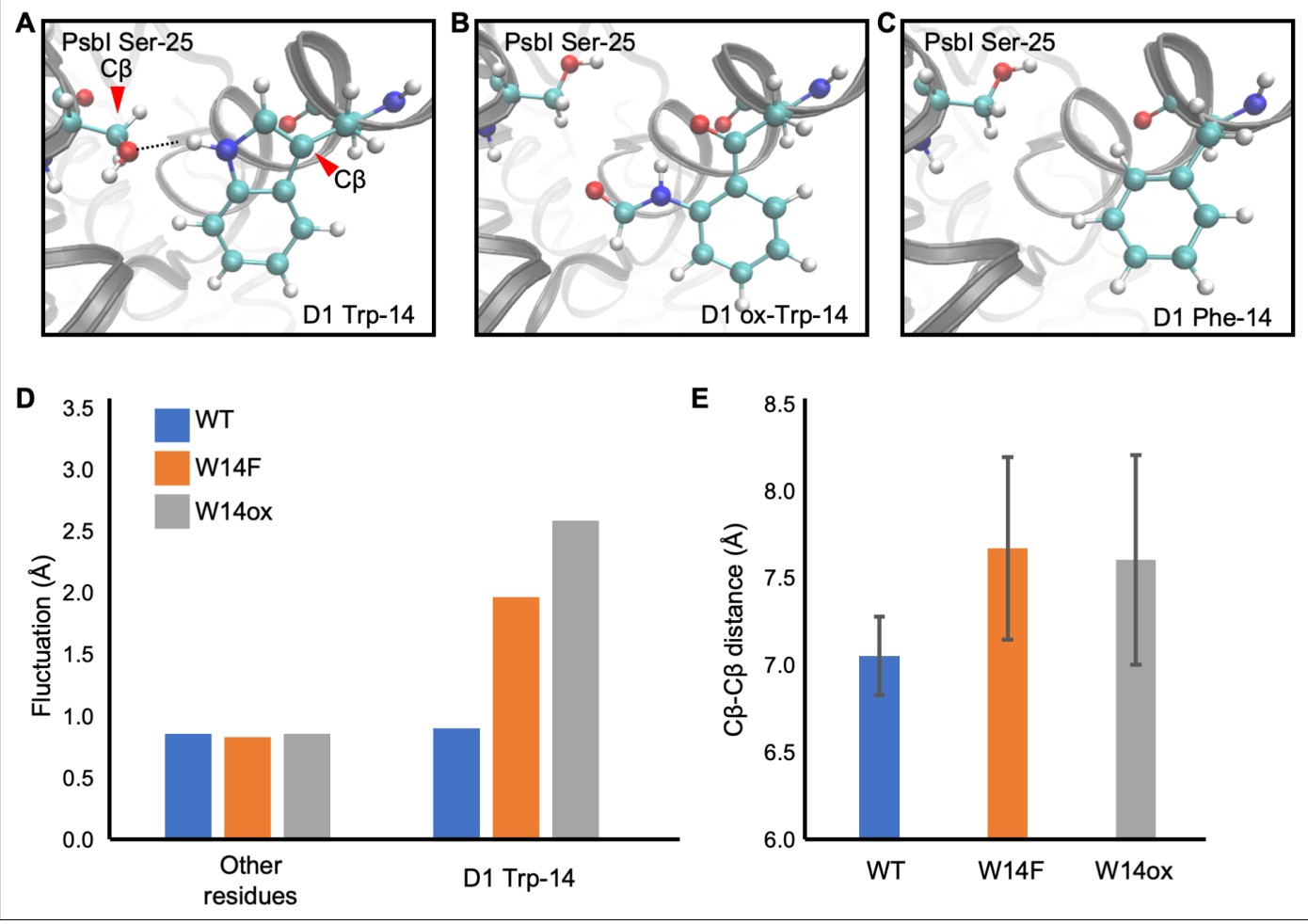

**Figure 7.** Snapshots and structural fluctuation of D1-Trp14 in molecular dynamics simulations of PSII. (**A**) The interaction between D1 Trp-14 and PsbI Ser-25. Dash line indicates the hydrogen bond between the side chains. (**B**) Position change of side-chain when D1 Trp-14 is oxidized to N-formylkynurenine. (**C**) Position change of side-chain when D1 Trp-14 is substituted to Phe. (**D**) The fluctuation of atoms at D1 Trp-14 in the MD simulation. (**E**) Averaged Cβ-Cβ distance between side chains of D1 Trp-14 and PsbI Ser-25. The error bars represent the standard deviations of the distances. The Cβ atoms are indicated as red arrowheads in **A**.

The online version of this article includes the following source data and figure supplement(s) for figure 7:

**Source data 1.** Source data for the fluctuation of atoms and Cβ-Cβ distance analysis in *Figure 7*.

**Figure supplement 1.** Atomic partial charges of NFK.

## Augmented interaction between D1 and FtsH by substituting Trp-14/317

Presented experimental results collectively raise the possibility that oxidation of Trp-14 is one of the key OPTMs for D1 degradation by FtsH. We raised a possibility that W14F mimics Trp-14 oxidation and shows increased FtsH association with D1. Since quantitative interaction of the protein and the protease remains to be elucidated, we performed differential pull-down assay. To emphasize the effect of the substituted amino acid residues and minimize potential oxidation of other amino acid residues, we decreased light intensity during cell culture and removed oxygen molecule from the buffer solution during the assay. *Chlamydomonas* cells were grown under dim light with gently shaking and were harvested at the mid-log phase. Subsequently, we isolated thylakoid membrane from the gently disrupted cells and performed co-immunoprecipitation in anoxic aqueous solution (*Figure 8A*). Quantification of D1 and D2 levels, normalized by FtsH in the co-immunoprecipitated sample showed that the relative D1 protein amounts was statistically higher in W14F/W317F than the control while D2 was

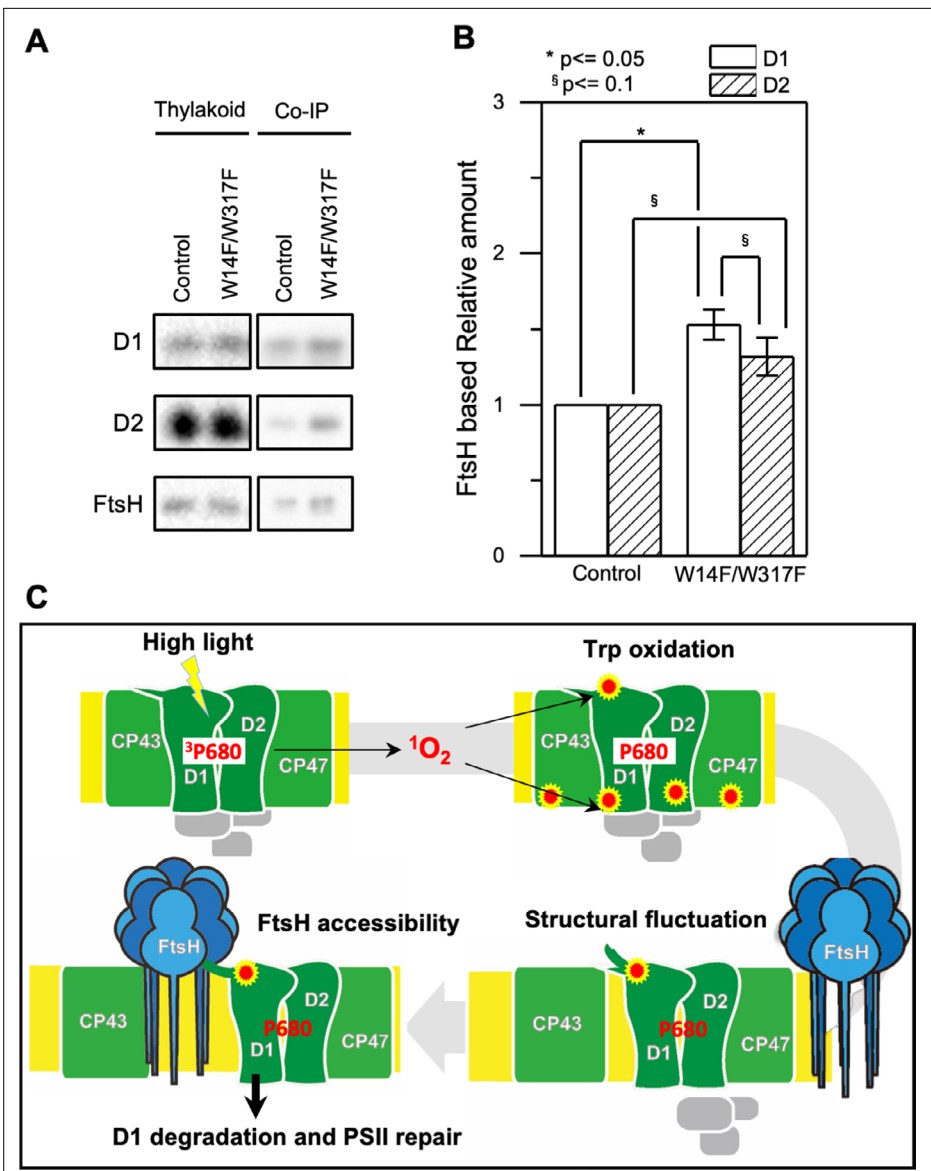

**Figure 8.** Augmented affinity of FtsH with D1 by W14F/W317F. (**A**) Coimmunoprecipitation was performed with anti-FtsH antibody using the thylakoid membrane isolated from control or D1-W14F/W317F. The polypeptides of thylakoid membrane or coimmunoprecipitated samples were separated by SDS-PAGE and detected by immunoblotting with anti-D1, anti-D2, and anti-FtsH antibody. (**B**) The immunoblotting signals are quantified and the ratio of D1 or D2 to FtsH are calculated. The averaged value and standard error for three biological replicates are shown. Significant difference was calculated by t-test and 0.1 (§) or 0.05 (*) probability confidence were indicated respectively. (**C**), A proposed model of photodamaged D1 recognition, in which Trp oxidation plays a role in recruiting FtsH. FtsH heterocomplexes (blue) and PSII core proteins (green) along with oxygen evolving protein complex (gray) in the thylakoid membrane are schematically shown. Trp-oxidized residues (red) are localized at both lumenal and stromal sides. Trp-14 located at the N-terminus alpha helix enhances association of FtsH, whose catalytic site faces stroma.

The online version of this article includes the following source data for figure 8:

**Source data 1.** Source data for western blot images and graphs in *Figure 8*.

fluctuating (*Figure 8B*). We concluded that W14F increased affinity between FtsH and the reaction center proteins, which leads to enhanced D1 degradation.

## Discussion

Recent progress in mass-spectrometry has advanced our understanding of holistic OPTM in photosynthetic protein complexes. Along this line, we investigated Trp oxidation in PSII in this study, and attempted to address whether any modification in amino acid residues was correlated with the PSII repair. PSII is one of the major sites for ROS generation due to photoinhibition, and oxidized amino acid residues in PSII core proteins have been reported previously (*Kale et al., 2017*; *Frankel et al., 2012*). In general, Met and Cys are sensitive amino acid residues for ROS-mediated oxidation (*Rinalducci et al., 2008*; *Ehrenshaft et al., 2015*). However, those OPTMs can be converted back in reduced forms by methionine sulfoxide reductase and disulfide reductase, respectively. In contrast, Trp oxidation is irreversible, and its replacement requires whole protein degradation and de novo synthesis, implicating Trp suitable for flagging photo-oxidative damaged proteins that undergo degradation in the PSII repair.

Previous studies have reported OPTM of several Trp residues in PSII core proteins in vitro (*Dreaden Kasson et al., 2012*; *Dreaden et al., 2011*; *Anderson et al., 2002*). In addition, Dogra et al. confirmed several oxidized Trp residues in PSII core proteins of *Arabidopsis* in vivo (*Dogra et al., 2019*). Our mass-spectrometry in *Chlamydomonas* further indicated that oxidation in some of the Trp residues was detected among algae and land plants. It is noteworthy that a majority of Trp residues are located on the lumenal side (*Table 2*) in the vicinity of the $Mn_4O_5Ca$ cluster, consistent with the previous observations that amino acid residues around the $Mn_4O_5Ca$ cluster are oxidized at the early stage of photoinhibition (*Kale et al., 2017*; *Frankel et al., 2012*). However, OPTM is not limited to the lumenal side but also found in the stromal side, which accounts for photoinhibition at the electron acceptor side (*Vass, 2012*). Our data indicated the presence of commonly oxidized Trp residues as represented by Trp-14 and Trp-317 in D1 protein. These are in fact Trp residues highly conserved among photosynthetic organisms (*Figure 1*). Technically, detection of light-dependent OPTM was considered to be difficult; an extensive OPTM leads to fully inactive PSII, whereas D1 degradation in the PSII repair may diminish OPTM under optimal light conditions. We therefore assumed that OPTM associated with PSII repair might accumulate significantly in the mutant that is defective in FtsH. Supporting this, our quantitative mass spectrometry indicated that *var2* lacking FtsH2 in *Arabidopsis* had increased levels of oxidation in Trp residues, particularly in Trp-14 and Trp-317. Although indirect, these results strongly suggested an interconnection between Trp oxidation and D1 degradation.

OPTM of Trp residues causes irreversible modification and is likely to mark photo-damaged D1 protein as a substrate for degradation. In the PSII repair, a series of events including migration of photo-damaged PSII to non-appressed regions of thylakoid membranes, release of CP43 from the PSII, and recognition of photo-damaged D1 for selective D1 degradation, are essential. In this scenario, FtsH interacts with a partially disassembled PSII complex lacking CP43 protein, called RC47 (*Kato and Sakamoto, 2009*; *Järvi et al., 2015*). Close access to the photo-damaged D1, followed by the recognition of its N-terminal region, is concomitantly necessary for FtsH to proceed with processive D1 degradation. Therefore, the OPTM would be involved in the CP43 disassembly or the recognition of damaged D1 protein. A recent study showed that exogenous ROS treatment leads to PSII disassembly supports this model (*McKenzie and Puthiyaveetil, 2023*). Krynická et al. indicate that the accessibility to PSII core proteins drives selective protein degradation by FtsH in the cyanobacterium *Synechocystis* PCC 6803 (*Krynická et al., 2015*). This observation suggests that D1 protein in the RC47 complex is promptly degraded even if D1 did not suffer from photodamage. On the other hand, all site-directed mutants mimicking Trp oxidation by Trp to Phe substitution have stable and functional PSII complexes under growth light, suggesting that the OPTM would not induce the disassembly of CP43. We also tested whether Trp oxidation in CP43 affects PSII repair by site-directed mutagenesis. Similarly to the case in D1, none of Trp substitutions at the site of OPTM in CP43 (Trp-353 and Trp-375) affected the D1 degradation. Additionally, the CP43 transformants did not show the increased photosensitivity (*Figure 3*). Our results somewhat appear to contradict the previous report in cyanobacterium that the mutants in which Trp-353 (Trp-352 in *Synechocystis* 6803) was substituted to Leu, Cys, or Ala increased photo-sensitivity under high-light conditions (*Anderson et al., 2002*). This might be due to the use of extremely high-light irradiation (5000 μmol photons $m^{-2}$ $s^{-1}$), under

which severe photo-damage in PSII complex was rendered. We consider that irreversible Trp oxidation in CP43, if to be repaired, may require a rapid turnover rate comparable to D1 degradation, which is not the case. Although further study is necessary to elucidate the disassembly mechanisms of CP43 during the PSII repair cycle, Trp oxidation in D1, rather than CP43 disassembly, might be important for the recognition of FtsH.

To examine its effect on D1 degradation, we performed site-directed mutagenesis of the corresponding Trp residues using *Chlamydomonas* chloroplast transformation. While Trp to Ala substitution in these sites (W14A or W317A) appeared to compromise PSII complex formation, Trp to Phe substitution (W14F and W317) gave us a hint in the critical role of OPTM. We showed that W14F, but not W317F, caused higher photo-sensitivity with the rapid decrease of D1 under high-light irradiation (*Figure 2*). Given that W14F affected neither D1 synthesis (*Figure 5*), stability of PSII complex formation, nor PSII activity under non-photoinhibitory conditions, it was concluded that the mutation results in enhanced D1 degradation. In our D1 degradation assay of wild type, generally, D1 turnover is too fast to detect unless inhibitor of chloroplast protein synthesis (CAM) is added. In sharp contrast, W14F proceeds with rapid D1 degradation even without CAM. Reportedly, numerous amino acid substitutions have been introduced in D1, which may or may not compromise PSII activity. To our knowledge, however, mutations that accelerate D1 degradation have not been found except for W14F in this study. We thus consider that Trp-14 is particularly important, at least for FtsH to recognize photodamaged D1 as described below.

Because FtsH-mediated D1 degradation is crucial for the PSII repair cycle, recognition of photodamaged D1 by FtsH protease is a critical step. Following observations suggest that Trp-14 oxidation is one of the key OPTMs for degrading photodamaged D1. First, enhanced D1 degradation in W14F well fits the notion that PTM in the N-terminus of D1 is important to execute processive degradation by FtsH, as proposed previously. For example, the lack of an N-terminal helix attenuates proper D1 degradation (*Komenda et al., 2007*; *Michoux et al., 2016*). The excision of N-terminal Met by organellar Met aminopeptidase and prokaryotic-like peptide deformylase was shown to be required for FtsH-mediated D1 degradation (*Adam et al., 2011*). Phosphorylation of the D1 N-terminus was also shown to affect proteolysis and contribute to the spatiotemporal regulation of D1 degradation pathway (*Koivuniemi et al., 1995*; *Rintamäki et al., 1996*; *Kato and Sakamoto, 2014*). Together with these, it is possible that Trp-14 oxidation is likely to play a role in 'photodamage-dependent' degradation, although we cannot rule out the possibility that other OPTMs may have additive effects. Second, our MD simulation strongly suggests that W14F is similar to Trp-14 oxidation W14* (modified as NFK) in terms of allowing a regional conformational change around the N-terminal helix, thereby increasing fluctuation of the side chain. This fluctuation appears to be manifested by losing hydrogen bonding with Ser-25 of PsbI, a short peptide localized close to D1 and CP43 in the PSII core complex. Although further studies are needed, our simulation is consistent with our notion that Trp-14 is a target of OPTM that alters subtle but critical structural change at the N-terminus of D1.

Based on these observations, we propose a working model of 'photodamaged D1 recognition' in which Trp oxidation plays a role in processive degradation by FtsH (*Figure 8C*). As a consequence of photoinhibition, ROS is produced around PSII and leads to OPTM of numerous residues. Among these, Trp-14 and Trp-317 are prone to oxidation likely due to their relative positions in PSII. While oxidation takes place in both, Trp-14 causes a conformational change at the N-terminus, which triggers enhanced access of FtsH for subsequent processive degradation. Supporting this, we observed augmented association between D1 and FtsH in W14F/W317F (*Figure 8*). It is unlikely, however, that Trp-14 oxidation alone is sufficient to drive degradation of photodamaged D1, because a stepwise dissociation of PSII core complexes is prerequisite. It is also possible that other oxidative modifications are synergistically involved in D1 degradation. A recent study using *Thermosynechococcus vulcanus* suggests that an oxidative modification of Phe at the DE-loop of the D1 protein disrupts the interaction between the PsbT and D1 (*Forsman and Eaton-Rye, 2021*). The stroma-exposed DE-loop of D1 is one of the possible cleavage sites by Deg protease, which facilitates the effective D1 degradation by FtsH in photoinhibitory conditions (*Kato et al., 2012*). Although W14 oxidation would be a key signal for D1 degradation by FtsH, other oxidative modifications in D1 could serve as a general degradation signal. Further study to directly monitor the oxidized D1 captured by FtsH will address these questions.

OPTM of Trp residues has been observed in various proteins (*Kasson and Barry, 2012*). For example, Trp oxidation has been identified in ATP synthase alpha subunit in mitochondria, one of the target proteins for oxidative stress in the mitochondrial inner membrane (*Rexroth et al., 2012*). In this case, the oxidation is not random but selectively targets specific Trp. The oxidized ATP synthase might be degraded by mitochondrial FtsH homologs, m-AAA and i-AAA proteases, which have an essential role in the quality control of aberrant proteins in mitochondrial membranes. Furthermore, a previous study in chloroplasts suggests the Trp oxidation in the stress responses related to singlet oxygen; *Dogra et al., 2019* report specific oxidation of Trp in EXECUTER1 (EX1), a sensor protein of singlet oxygen in plastid signaling. The oxidation of specific Trp residue is required for ROS signaling mediated by light-dependent EX1 degradation by FtsH (*Dogra et al., 2019*). Together with our results, these reports imply a general mechanism between oxidized modification of target protein and substrate recognition by FtsH. Future proteomic approaches for investigating OPTM will reveal the general substrate recognition mechanisms by FtsH in the thylakoid membranes.

## Methods

### Detection of Trp oxidation in *Arabidopsis*

Chloroplasts were isolated from 3-week-old plants of WT and *var2* (SAIL_253_A03) grown under continuous light (80 μmol photons $m^{-2}s^{-1}$ at 20 ± 2°C) conditions. The collected rosette leaves were homogenated in chloroplast isolation buffer [50 mM Hepes-KOH pH 8, 5 mM $MgCl_2$, 5 mM EDTA pH 8, 5 mM EGTA pH 8, 10 mM $NaHCO_3$, and 0.33 M D-sorbitol, supplemented with SIGMAFAST Protease Inhibitor (1 tablet per 100 ml)]. The homogenate was filtered through four layers of Miracloth and centrifuged at 400×*g* for 8 min at 4 °C. The pellets were suspended in isolation buffer and loaded onto a two-step Percoll gradient (40:80%) solution to separate intact and broken chloroplasts. The intact chloroplasts enriched between the two Percoll steps were carefully collected and washed twice with HS buffer (50 mM Hepes-KOH pH 8, and 0.33 M D-sorbitol). Chloroplasts corresponding to equal amounts of chlorophyll were lysed, and the proteins extracted using 6 M guanidine hydrochloride buffer (guanidine hydrochloride dissolved in 100 mM Tris, pH 8.5). The lysed samples were sonicated in an ice bath for 1 min with a pulse of 3 s 'on' and 5 s 'off', followed by heating at 95 °C for 5 min, and then centrifugation at 21 000 g for 30 min at 4 °C. Total protein content was estimated using a PierceTM BCA protein assay kit (Thermo Fisher Scientific).

Mass spectrometric analysis for protein identification and PTM analysis was done according to our previous study (*Dogra et al., 2019*). For MS analysis, equal amounts of total protein (2 μg $μl^{-1}$) from three independent biological samples were denatured using 10 mM DTT at 56 °C for 30 min followed by alkylation in 50 mM iodoacetamide at room temperature for 40 min in the dark. Reduced-alkylated proteins were then desalted a Nanosep membrane (Pall Corporation, MWCO 10 K) in 200 μL of 100 mM $NH_4HCO_3$ buffer, followed by digestion in buffer containing 40 ng/μl trypsin in 100 mM $NH_4HCO_3$ (corresponding to the enzyme-to-protein ratio of 1:50) at 37 °C for 20 h. The digested peptides were dried and resuspended in 0.1% (v/v) formic acid solution. Digested peptides were separated using nanoAcquity Ultra Performance LC (Waters, Milford, MA, USA) and analyzed by using Q Exactive Mass Spectrometer (Thermo Fisher Scientific, San Jose, CA, USA) as described in our previous study (*Dogra et al., 2019*). The mass spectra were submitted to the Mascot Server (version 2.5.1, Matrix Science, London, UK) for peptide identification and scanned against the *Arabidopsis* protein sequences (downloaded from TAIR; http://www.arabidopsis.org/). Database searches were carried out with peptide mass tolerance of 20 ppm, fragment mass tolerance of 0.02 Da, and a maximum of two missed cleavages. Carbamidomethylation of Cys was set as a fixed modification, while oxidations of Met and Trp were defined as variable modifications. The significance threshold for search results was set at a p-value of 0.05 and an Ions score cut-off of 15. For quantification, raw MS data files were processed and analyzed using MaxQuant software (version 1.5.8.3) with a label-free quantitation (LFQ) algorithm. Parent ion and MS2 spectra were searched against the *Arabidopsis* protein sequences. The precursor ion tolerance was set at 7 ppm with an allowed fragment mass deviation of 20 ppm. Carbamidomethylation of Cys was set as a fixed modification, while oxidations of Met and Trp were defined as variable modifications. Peptides with a minimum of six amino acids and a maximum of two missed cleavages were allowed. False discovery rate (FDR) was set to 0.01 for both peptide and protein identification. The absolute intensity values were used to calculate the

abundance of oxidized peptides. Label-free quantitation of oxidized peptides using mass spectrometry were performed according to previously described method (*Luber et al., 2010*; *Schwanhäusser et al., 2011*; *Duan et al., 2019*).

## Strains and generation of chloroplast transformants in *Chlamydomonas*

The *psbA* deletion mutant of the green alga *Chlamydomonas reinhardtii*, Fud7 (*Bennoun et al., 1986*) was used for chloroplast transformation in this study. The vector, which lacks large portion of *psbA* gene (*Takahashi et al., 1996*), was used for plasmid construction. To obtain *psbA* mutants, each *psbA* transformation vector was biolistically delivered into chloroplast of the Fud7 mutant using a particle gun (IDERA GIE-III, TANAKA Co. Ltd., Sapporo, Japan). Chloroplast transformants were selected by at least four rounds of single colony purification on TAP agar plates containing spectinomycin (150 µg mL$^{-1}$) as described previously (*Takahashi et al., 1996*). The CP43 mutants were generated according to *Kuroda et al., 2021*. The psbD gene in the Fud7 was disrupted and we obtained Fud7-ΔpsbD mutant as a recipient for CP43 mutagenesis. The DNA delivery methods are the same with *psbA* mutagenesis experiment.

## Detection of Trp oxidation in *Chlamydomonas*

Cultivation of the algae was carried out under constant light (50 µmol photons m$^{-2}$s$^{-1}$ or 500 µmol photons m$^{-2}$s$^{-1}$) in TAP medium for 24 h. Cells were harvested by centrifugation (2500 x g for 5 min at room temperature), frozen in liquid nitrogen and stored at –80 °C until further use. For protein extraction, lysis buffer (100 mM Tris/HCl pH 8.5, 2% (w/v) SDS, 1 mM PMSF, 1 mM benzamidine) was added to frozen cell pellets and incubated for 10 min at 65 °C and 1000 rpm in a Thermomixer (Eppendorf, Germany). The lysate was cleared by centrifugation (18,000 x *g* for 10 min at 25 °C) and the protein content of the supernatant was determined using the Pierce BCA protein assay kit (Thermo Fisher Scientific). Reduction, alkylation and tryptic digestion (50 µg of protein per sample) was performed in centrifugal filters (Amicon Ultra-0.5, 30 kDa cut-off, Merck Millipore) according to the FASP protocol (*Wiśniewski et al., 2009*). Peptides (5 µg per sample) were desalted using self-packed C18-StageTips as previously described (*Kulak et al., 2014*), followed by vacuum centrifugation until dry. Prior to LC-MS/MS analysis peptide samples were resuspended in 2% (v/v) acetonitrile/0.05% (v/v) trifluoroacetic acid at a concentration at a concentration of 1 µg/µl. LC-MS/MS analysis was carried out using an Ultimate 3000 nanoLC (Thermo Fisher Scientific) coupled to an Q Exactive Plus mass spectrometer (Thermo Fisher Scientific) via a nanospray interface. Samples (1 µl) were loaded on a trap column (C18, Acclaim PepMap 100, 300 µM×5 mm, 5 µm particle size, 100 Å pore size; Thermo Scientific) at a flow rate of 10 µl/min for 3 min using 2% (v/v) acetonitrile/0.05% (v/v) trifluoroacetic acid in ultrapure water. Subsequently, peptide separation was performed on a reversed phase column (C18, Acclaim Pepmap C18, 75 µm x 50 cm, 2 µm particle size, 100 Å pore size, Thermo Fisher Scientific) at a flow rate of 250 nl/min using the eluents 0.1% (v/v) formic acid in ultrapure water (A) and 80% (v/v) acetonitrile/0.1% (v/v) formic acid in ultrapure water (B). The following gradient was applied: 2.5–5% B over 10 min, 5–22% B over 90 min, 22–30% B over 70 min, 30–99%B over 10 min, 99% B for 20 min.

MS full scans (m/z 350–1600) were acquired in positive ion mode at a resolution of 70,000 (FWHM, at m/z 200) with internal lock mass calibration on m/z 445.120025. The AGC target was set to 3e6 and the maximum injection time to 50ms. For MS2, the 12 most intense ions with charge states 2–4 were fragmented by higher-energy c-trap dissociation (HCD) at 27% normalized collision energy. AGC target value was set to 5e4, minimum AGC target to 5.5e2, maximum injection time to 55ms and precursor isolation window to 1.5 m/z.

Peptide and protein identification were carried out in Proteome Discoverer 2.4 (Thermo Fisher Scientific) using the MSFragger node (MSFragger 3.0)(*Kong et al., 2017*) with default parameters for closed searches (precursor mass tolerance: 50 ppm, precursor true tolerance: 20 ppm, fragment mass tolerance: 20 ppm, maximum missed cleavages: 1). Spectra were searched against a concatenated sequence database containing nucleus-encoded proteins (https://www.phytozome.org, assembly version 5.0, annotation version 5.6), supplemented with proteins encoded in the chloroplast (NCBI BK000554.2) and mitochondria (NCBI NC_001638.1), as well as common contaminants (cRAP, https://www.thegpm.org/crap/). Carbamidomethylation was set as static modification. The following variable modifications were defined: N-acetylation of protein N-termini, oxidation of methionine, and various products of tryptophan oxidation kynurenine (+3.995 Da), hydroxytryptophan (+15.995 Da),

hydroxykynurenine (+19.990 Da), N-formylkynurenine (+31.990 Da), dihydroxy-N-formylkynurenine (+63.980 Da). Peptide-spectrum-matches (PSMs) were filtered using the Percolator node to satisfy a false discovery rate (FDR) of 0.01. Subsequently, identifications were filtered to achieve a peptide and protein level FDR of 0.01.

## Growth test

Cells were grown in TAP liquid medium without shaking at 23–24°C under the light-dark synchronized condition (10 hr light at 50 µmol photons m$^{-2}$s$^{-1}$ or less and 14 hr darkness). Subsequently the cells were harvested by centrifugation at 2000×$g$ for 10 min at 25 °C and were suspended in TP (Tris Phosphate) medium for washing. After finishing the washing process, the cell concentration was adjusted at 25 ng Chl µL$^{-1}$ with TP medium. The liquid culture was spotted on solid medium at 100 ng Chlorophylls/spot. When we evaluate the cellular growth rate in the liquid culture, the cells grown under 30 µmol photons m$^{-2}$s$^{-1}$ in TAP medium were suspended in the TP medium at 0.1 of OD750 and were incubated under 30 or 350 µmol photons m$^{-2}$s$^{-1}$.

## Measurement of photosynthetic activity

Chlorophyll fluorescence induction kinetics of *Chlamydomonas* transformants were measured using a pulse amplitude-modulated fluorometer (Dual-PAM-100; Heinz Walz GmbH). Before measurements, cultured cells were maintained in the dark for 5 min to oxidize the plastoquinone pool fully. Initial fluorescence yield of PSII ($F_O$) and maximal fluorescence yield of PSII ($F_M$) were measured. Maximal PSII quantum yield ($F_V/F_M$) was determined as $F_V/F_M = (F_M–F_O)/F_M$. Light-induced oxygen-evolving activity of cells was measured using a Clark-type $O_2$ electrode (Oxytherm OXYT1; Hansatech Instruments). Briefly, cells were grown in TAP culture under 5 µmol photons m$^{-2}$ s$^{-1}$ to reach 5–10 µg Chl mL$^{-1}$. $O_2$-evolving activity of cells (5 µg Chl mL$^{-1}$) in the presence of 0.3 mM 2,6-dichloro-1,4-benzoquinone was measured using a Clarke-type $O_2$ electrode with an actinic light at 7800 µmol photons m$^{-2}$ s$^{-1}$ at 25 °C as described (*Kuroda et al., 2014*).

## Immunoblotting

Total proteins were solubilized in SDS-PAGE sample (125 mM Tris-HCl, pH 6.8, 2% [w/v] SDS, 100 mM dithiothreitol, 10% [v/v] Glycerol, 0.05% [w/v] BPB) buffer at 96 °C for 1 min, and then were loaded based on equal chlorophyll. The proteins were electrophoretically transferred onto polyvinylidene difluoride membrane (Atto Corp.) after SDS-PAGE. The membranes were incubated with specific polyclonal antibodies: anti-D1 (raised against N-termimus, dilution 1:5000 *Kato et al., 2012*), anti-D2 (AS06 146, Agrisera; dilution, 1:5000), anti-CP43 (AS11 1787, Agrisera; dilution, 1:5000), anti-PsaA (a gift from Kevin Redding, Arizona State University, dilution 1:5000), and anti-Lhca1, dilution 1:5000; (*Ozawa et al., 2018*). The signals were visualized by using a Luminata Forte Western HRP Substrate (Merck Millipore) with Molecular Imager ChemiDoc XRS +imaging system (Bio Rad Laboratories, Inc, USA). Signal intensities were quantified using NIH Image.

## D1 degradation assay

Cells were grown in TAP liquid medium at 22 °C under continuous light-condition (30 µmol photons m$^{-2}$s$^{-1}$). Cultured cells were harvested by centrifugation at 600×$g$ for 5 min. The cell pellets were resuspended in a new TAP liquid medium as a final concentration of 0.5 µg Chl mL$^{-1}$. Then, the cells were preincubated in the presence or absence of chloramphenicol (100 µg mL$^{-1}$) in the dark for 30 min. Subsequently, the cells were incubated under high-light or growth light conditions (350 or 30 µmol photons m$^{-2}$s$^{-1}$) with stirring. Cells in 400 µl culture were collected at each time points (30, 60, 90 min) by centrifugation, and the resulting cell pellet was resuspended in 100 µl of SDS-PAGE sample buffer.

## Pulse labeling of chloroplastic proteins

Cells grown in pre-culture medium (TAP media with less sulfur) were harvested by centrifugation at 600×$g$ for 5 min and were washed by TAP media containing no sulfur. After centrifugation, the cells were resuspended to 25 µg Chl mL$^{-1}$ in TAP media containing no sulfur and incubated for 2 h. Subsequently, sulfur-starved cells were labeled with 5 µ Ci mL$^{-1}$ [$^{35}$S]Na$_2$SO$_4$ (American Radiolabeled Chemicals) in the light at 50 µmol photons m$^{-2}$s$^{-1}$ in the presence of 10 µg mL$^{-1}$ cycloheximide. At each time point (1, 2, 4 min), cell samples were collected and immediately frozen in liquid nitrogen.

## Thylakoid membrane isolation and the following co-immunoprecipitation in anoxic aqueous solution

Cells grown in TAP medium under 5 μmol photons m$^{-2}$ s$^{-1}$ were harvested by centrifugation at 2,000×g for 10 min at 25 °C. All buffers were incubated at 25 °C for 60 min in the presence of 100 mM glucose, 40 U/mL glucose oxidase, and 50 U/mL catalase to remove oxygen before chilling. Cells were suspended in suspension buffer (10 mM HEPES-KOH pH 8.0), broken by double passage through an airbrush at a pressure of 0.2 MPa (0.2 mm aperture airbrush). The broken materials were suspended in high sucrose concentration solution (1.8 M sucrose, 10 mM HEPES-KOH pH 8.0), and then a low sucrose concentration solution (1.0 M sucrose, 10 mM HEPES-KOH pH 8.0) and suspension buffer were layered in this order. Thylakoid membrane was floated at the interface between high-sucrose concentration solution and low sucrose concentration solution after centrifugation (at 20,000×g, for 60 min, at 25 °C). The recovered thylakoid membrane was suspended in the suspension buffer.

The anti-VAR2 antibody (*Sakamoto et al., 2003*) was conjugated with magnetic beads (Magnosphere, MS 160/Tosyl, JSR life sciences, Japan) by the presence of the fully chemically synthesized polymer (Blockmaster CE210, JSR life sciences, Japan) according to the instruction manual. The conjugated and blocked magnetic beads were suspended in the suspension buffer (10 mM HEPES-KOH pH 8.0) after washing TBS-T. Prior to the incubation with solubilized thylakoid membrane, the magnetic beads were resuspended in the suspension buffer of which oxygen was removed enzymatically by incubating at 25 °C for 60 min in the presence of 100 mM glucose, 40 U/mL glucose oxidase, and 50 U/mL catalase.

Thylakoid membrane was solubilized sequentially; thylakoid membrane (1.0 mg Chlorophyll/mL) was incubated with 1.0% (w/v) glyco-diosgenin (GDN) and subsequently n-dodecyl-α-maltoside was added at 1.0% (w/v), and finally the mixture was diluted at twice volume with suspension buffer. The solubilized material was incubated with FtsH conjugated magnetic beads for 60 min at 4 °C after removal of debris by centrifugation (at 20,000×g, for 1 min, at 4 °C). The beads were washed six times with suspension buffer containing 0.02% (w/v) GDN and were incubated with elution buffer (125 mM Tris-HCl pH 6.8, 2% (w/v) Lithium Dodecyl sulfate, 0.1% (w/v) Sodium Dodecyl sulfate, and 25% (w/v) glycerol) for 60 min on ice. The eluted sample was directly loaded on individual sample slot on SDS-PAGE to separate polypeptides. All buffers except elution buffer were incubated at 25 °C for 60 min in the presence of 100 mM glucose, 40 U/mL glucose oxidase, and 50 U/mL catalase to remove oxygen before chilling.

## Molecular dynamics simulations of D1 N-term in PSII complex

The MD simulations for PSII were performed using the X-ray crystal structure determined at 1.9 Å resolution (PDB: 3ARC)(*Umena et al., 2011*) and based on the same procedure described previously (*Sakashita et al., 2017b*; *Sakashita et al., 2017a*; *Kawashima et al., 2018*), except for the following points. To investigate the structural fluctuation of the N terminal region of the D1 subunit, we restructured the N-terminal region between D1-Met1 and D1-Ser10 that was lacking in the crystal structure, using MOE program (2018). After structural optimization with positional restraints on heavy atoms of the PSII assembly, the system was heated from 0.001 to 300 K over 5.0 ps, with a 0.05-fs time step. The positional restraints on heavy atoms were gradually released over 16.5 ns. After an equilibrating MD run for 40 ns, a production run was conducted over 495 ns with an MD time step of 1.5 fs. The SHAKE algorithm was used for hydrogen constraints (*Ryckaert et al., 1977*). The structure of the D1-W14F mutant was modeled from the crystal structure of WT. The MD simulations were based on the AMBER-ff14SB force field for protein residues and lipids (*Maier et al., 2015*). The water molecules were described by TIP3P model (*Jorgensen et al., 1983*). For NFK, we employed the generalized Amber force field (GAFF) parameter set (*Wang et al., 2004*). The atomic partial charges of NFK were determined by fitting the electrostatic potential by using the RESP procedure (*Bayly et al., 1993*) (for calculated charges, see *Figure 7—figure supplement 1*). The electronic wave functions were calculated after geometry optimization with the density functional theory of the B3LYP/6–31 G** level by using JAGUAR (ver. 8.0, https://www.schrodinger.com/products/jaguar). MD simulations were conducted using the MD engine NAMD (*Phillips et al., 2005*). The atomic fluctuation was calculated as the root mean square fluctuation (RMSF) of heavy atoms from the averaged structure of PSII over the whole MD trajectory.

## Plasmid construction

To construct *aadA* marker for *psbA* transformation vectors, a 483 bp DNA fragment was amplified by PCR from plasmid pLM20 (*Michelet et al., 2011*) using primers On#1 (HK1174) and On#2 (HK1175), digested by XhoI and EcoRI and ligated into pBluescript KS⁻ to generate plasmid pSXY501. An XhoI/Bsp119I-digested fragment from the pSXY501 and a ClaI/SpeI-digested fragment from pUC-atpX-AAD (*Goldschmidt-Clermont, 1991*) were simultaneously ligated into XhoI/NheI site of the pSXY501 to generate pSXY503. An NsiI/EcoRI-digested fragment from the pSXY503 was ligated into PstI/EcoRI-digested pBluescript KS⁻ to generate pSXY504. Annealed oligo DNAs (On#3 and On#4) was ligated into HindIII/XbaI-digested pUC18 to generate pSRE6. An NheI/EcoRV-digested fragment from the pSXY504 was ligated into XbaI/EcoRV-digested pSRE6 to generate pSXY518.

A chloroplast transformation vector for *psbA* mutagenesis was generated by inserting Mva1269I/ApaI-digested fragment from a plasmid, which contains *psbA* exon 5 and 3' flanking region between EcoRI and XhoI sites, into Mva1269I/ApaI-digested pR12-EX-50-AAD (*Takahashi et al., 1996*) to generate pSXY235. A BamHI/BglII fragment from pSXY518 was inserted into BamHI-digested pSXY235 to generate pSXY236A, in which *psbA* and *aadA* were transcribed in the same orientation. Orientation of the *aadA* was confirmed by restriction enzyme digestion.

To introduce a point mutation at the D1-W14 codon, an EcoRI/NspI fragment from plasmid pR12-EX-50-AAD, which contains *psbA* exon 1, was cloned into pBluescript KS⁻ to generate pSXY232. A PCR-amplified DNA fragment from AflII/Bpu10I-digested pSXY232 using primers On#5 and On#6 was inserted into NdeI-digested pSXY232 using In-Fusion HD cloning kit to generate pSXY233. A DNA fragment that contains D1-W14A codon was amplified from total *Chlamydomonas* DNA by PCR using primers On#7 and On#8. A DNA fragment that contains D1-W14F codon was amplified from total *Chlamydomonas* DNA by PCR using primers On#7 and On#9. Each DNA fragment that contains mutated D1-W14A codon or D1-W14F codon was inserted into AfeI-digested pSXY233 using In-Fusion HD cloning kit to generate pSXY237 and pSXY238, respectively. An AflII/NsiI-digested fragment from pSXY237 and pSXY238 was ligated into AflII/NsiI-digested pSXY236A to generate transformation vectors pSXY239A (D1-W14A) and pSXY240A (D1-W14F), respectively. The DNA segments amplified by PCR was confirmed by DNA sequencing using a primer On#10.

To substitute D1-W317 codon by Alanine (A) codon, a PCR-amplified fragment from total *Chlamydomonas* DNA using primers On#11 and On#12 and a PCR-amplified fragment from total *Chlamydomonas* DNA using primers On#13 and On#14 were simultaneously inserted into Mva1269I/SexAI-digested pSXY236A to generate pSXY241A (D1-W317A). A transformation vector pSXY243A for D1-W14A/D1-W317A double mutant was constructed by inserting AflII/NsiI-digested fragment into AflII/NsiI site of the pSXY241A. To substitute D1-W317 codon by Phenylalanine (F) codon, a PCR-amplified fragment from total *Chlamydomonas* DNA using primers On#11 and On#15 and a PCR-amplified fragment from total *Chlamydomonas* DNA using primers On#13 and On#14 were simultaneously inserted into Mva1269I/SexAI-digested pSXY236A to generate pSXY242A (D1-W317F). A transformation vector pSXY244A for D1-W14F/D1-W317F double mutant was constructed by inserting AflII/NsiI-digested fragment into AflII/NsiI site of the pSXY242A. The DNA segments amplified by PCR was confirmed by DNA sequencing using a primer On#16.

To generate CP43 mutants, we first deleted *psbC* from chloroplast genome of the Fud7. Plasmid P-578 obtained from Chlamydomonas Resource Center (University of Minnesota, St. Paul, MN) was digested with EcoRI and SnaBI, and cloned into the EcoRI-HincII site of pBluescript KS- to generate pSXY1001. Plasmid pSXY1001 was digested with EcoRI and HincII, and cloned into the EcoRI-SwaI site of pBluescript KS- to generate pSXY1002. Two PCR products were amplified from the pSXY1002 using primers On#18/On#19 or On#17/On#20, digested with EcoRI or PmlI and then simultaneously inserted into the EcoRI-PmlI site of the pSXY1002 to generate pSXY1003. An PCR product was amplified from pSXY1001 using primers On#21 and On#22, digested with PmlI and cloned into the SwaI-PmlI site of the pSXY1003 to generate pSXY1004. An PCR product was amplified from the Fud7 total DNA using primers On#19 and On#23, digested with SwaI and EcoRI, and cloned into the SmaI-EcoRI site of the pSXY1004 to generate pSXY1005. A BamHI-EcoRI fragment of the pSXY1005 was cloned into the BamHI-EcoRI site of the pSXY1002 to generate pSXY1006. An PCR product was amplified from wild type Chlamydomonas total DNA using primers On#24 and On#25, digested with SacI and PacI, and cloned into the SacI-PacI site of the pSXY1006 to generate pSXY1007. To construct a recyclable aadA marker, an NsiI-BamHI fragment of the pSXY503 (Kuroda2021) was cloned into the

PstI-BamHI site of pUC18 to generate pSXY506. A HindIII-SmaI fragment of the pSXY506 was cloned into the HindIII-PmlI site of the pSXY1007 to generate transformation vector pSXY1008 for deletion of the *psbC* gene.

To introduce a point mutation(s) at the CP43-W365 and the CP43-W387 codons, an inverse PCR product was amplified from the pSXY1002 using primers On#26 and On#27, and self-ligated to generate pSXY1032. An EcoRI-PmlI fragment of the pSXY1032 was cloned into the EcoRI-PmlI site of the pSXY1007 to generate pSXY1033. For CP43-W365 and CP43-W387 mutagenesis, each PCR product, which carried mutated CP43-W365 and/or mutated CP43-W387 codons, was amplified from wild type Chlamydomonas total DNA using primers On#28 to On#33, and introduced into PstI site of the pSXY1033 with In-Fusion HD cloning kit (Takara) to generate transformation vectors pSXY1034 to pSXY1039 (Seq: On#22).

## Acknowledgements

We thank Rie Hijiya and Tsuneaki Takami for their technical assistance. We also thank Dr. Michel Goldschmidt-Clermont (University of Geneva) for providing Plasmid pLM20. This work was supported by the following KAKENHI grants; 23H04959 from the Ministry of Education, Culture, Sports, Science and Technology (MEXT) to WS; 21H02508 from the Japan Society for the Promotion of Science (JSPS) to WS; 21K06221 from JSPS to YK; 23H04963 from MEXT to KS; 20H03217 and 23H02444 from JSPS to HI; the Oohara Foundation to WS; CNRS and Sorbonne Université (basic support to UMR7141) and the Initiative d'Excellence program (grant DYNAMO, Agence Nationale de la Recherche ANR-11-LABX-0011–01) to CdV.

## Additional information

### Funding

| Funder | Grant reference number | Author |
|---|---|---|
| Ministry of Education, Culture, Sports, Science and Technology | KAKENHI (23H04959) | Wataru Sakamoto |
| Japan Society for the Promotion of Science | KAKENHI (21H02508) | Wataru Sakamoto |
| Japan Society for the Promotion of Science | KAKENHI (21K06221) | Yusuke Kato |
| Ministry of Education, Culture, Sports, Science and Technology | KAKENHI (23H04963) | Keisuke Saito |
| Oohara Foundation | | Wataru Sakamoto |
| Sorbonne Université | UMR7141 | Catherine de Vitry |
| Agence Nationale de la Recherche | ANR-11-LABX-0011-01 | Catherine de Vitry |
| Japan Society for the Promotion of Science | KAKENHI (20H03217) | Hiroshi Ishikita |
| Japan Society for the Promotion of Science | KAKENHI (23H02444) | Hiroshi Ishikita |

The funders had no role in study design, data collection and interpretation, or the decision to submit the work for publication.

### Author contributions

Yusuke Kato, Data curation, Validation, Investigation, Writing – original draft; Hiroshi Kuroda, Shin-Ichiro Ozawa, Keisuke Saito, Vivek Dogra, Data curation, Validation, Investigation; Martin Scholz, Data curation, Investigation; Guoxian Zhang, Investigation; Catherine de Vitry, Resources, Investigation; Hiroshi Ishikita, Data curation, Validation; Chanhong Kim, Validation, Investigation; Michael Hippler,

Yuichiro Takahashi, Supervision; Wataru Sakamoto, Conceptualization, Supervision, Validation, Writing – original draft, Writing - review and editing

### Author ORCIDs
Yusuke Kato http://orcid.org/0000-0003-1718-9121
Shin-Ichiro Ozawa http://orcid.org/0000-0001-7698-5350
Keisuke Saito http://orcid.org/0000-0002-2293-9743
Vivek Dogra https://orcid.org/0000-0003-1853-8274
Hiroshi Ishikita http://orcid.org/0000-0002-5849-8150
Chanhong Kim http://orcid.org/0000-0003-4133-9070
Michael Hippler http://orcid.org/0000-0001-9670-6101
Wataru Sakamoto http://orcid.org/0000-0001-9747-5042

Reviewer #1 (Public Review): https://doi.org/10.7554/eLife.88822.3.sa1
Reviewer #2 (Public Review): https://doi.org/10.7554/eLife.88822.3.sa2
Reviewer #3 (Public Review): https://doi.org/10.7554/eLife.88822.3.sa3
Author Response https://doi.org/10.7554/eLife.88822.3.sa4

---

## Additional files

### Supplementary files
• MDAR checklist

### Data availability
All data generated or analyzed during this study are included in the manuscript and supporting files. Source data files have been provided for Figures 1-8.

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

# Appendix 1

**Appendix 1—key resources table**

| Reagent type (species) or resource | Designation | Source or reference | Identifiers | Additional information |
|---|---|---|---|---|
| Strain, strain background (*A. thaliana*) | *var2* | NASC | SAIL_253_A03 | |
| Strain, strain background (*C. reinhardtii*) | *Fud7* | PMID: 24307274 | | |
| Strain, strain background (*C. reinhardtii*) | *ΔpsbC/Fud7* | PMID: 8676881 | | |
| Strain, strain background (*C. reinhardtii*) | *D1-W14A* | This paper | | Line stocked in the Sakamoto lab |
| Strain, strain background (*C. reinhardtii*) | *D1-W317A* | This paper | | Line stocked in the Sakamoto lab |
| Strain, strain background (*C. reinhardtii*) | *D1-W14A-W317A* | This paper | | Line stocked in the Sakamoto lab |
| Strain, strain background (*C. reinhardtii*) | *D1-W14F* | This paper | | Line stocked in the Sakamoto lab |
| Strain, strain background (*C. reinhardtii*) | *D1-W317F* | This paper | | Line stocked in the Sakamoto lab |
| Strain, strain background (*C. reinhardtii*) | *D1-W14A-W317F* | This paper | | Line stocked in the Sakamoto lab |
| Strain, strain background (*C. reinhardtii*) | *CP43-W353A* | This paper | | Line stocked in the Sakamoto lab |
| Strain, strain background (*C. reinhardtii*) | *CP43-W375A* | This paper | | Line stocked in the Sakamoto lab |
| Strain, strain background (*C. reinhardtii*) | *CP43-W353A-W375A* | This paper | | Line stocked in the Sakamoto lab |
| Strain, strain background (*C. reinhardtii*) | *CP43-W353F* | This paper | | Line stocked in the Sakamoto lab |
| Strain, strain background (*C. reinhardtii*) | *CP43-W375F* | This paper | | Line stocked in the Sakamoto lab |
| Strain, strain background (*C. reinhardtii*) | *CP43-W353F-W375F* | This paper | | Line stocked in the Sakamoto lab |
| Strain, strain background (*C. reinhardtii*) | *ftsh1-1* | PMID: 24449688 | | |

*Appendix 1 Continued on next page*

*Appendix 1 Continued*

| Reagent type (species) or resource | Designation | Source or reference | Identifiers | Additional information |
|---|---|---|---|---|
| Strain, strain background (*C. reinhardtii*) | D1-W14F ftsH | This paper | | Line stocked in the Sakamoto lab |
| Strain, strain background (*C. reinhardtii*) | D1-W14F-W317F ftsH | This paper | | Line stocked in the Sakamoto lab |
| Antibody | anit-D1 (rabbit polyclonal) | PMID: 22698923 | | (1:5000) |
| Antibody | anit-D2 (rabbit polyclonal) | Agrisera | AS06 146 | (1:5000) |
| Antibody | anit-CP43 (rabbit polyclonal) | Agrisera | AS11 1787 | (1:5000) |
| Antibody | anit-PsaA (rabbit polyclonal) | a gift from Kevin Redding, Arizona State University | | (1:5000) |
| Antibody | anti-Lhca1 (rabbit polyclonal) | PMID: 30126869 | | (1:5000) |
| Antibody | anit-FtsH (rabbit polyclonal) | PMID: 14630971 | | (1:5000) |
| Recombinant DNA reagent | pLM20 | PMID: 20809927 | | |
| Recombinant DNA reagent | pR12-EX-50-AAD | PMID: 8665094 | | |
| Recombinant DNA reagent | pUC-atpX-AAD | PMID: 1651475 | | |
| Recombinant DNA reagent | pSXY239A (D1-W14A) | This paper | | See "Plasmid construction" in Methods |
| Recombinant DNA reagent | pSXY240A (D1-W14F) | This paper | | See "Plasmid construction" in Methods |
| Recombinant DNA reagent | pSXY241A (D1-W317A) | This paper | | See "Plasmid construction" in Methods |
| Recombinant DNA reagent | pSXY242A (D1-W317F) | This paper | | See "Plasmid construction" in Methods |
| Recombinant DNA reagent | pSXY243A (D1-W14A-W317A) | This paper | | See "Plasmid construction" in Methods |
| Recombinant DNA reagent | pSXY244A (D1-W14F-W317F) | This paper | | See "Plasmid construction" in Methods |
| Recombinant DNA reagent | pSXY1034 (CP43-W353A) | This paper | | See "Plasmid construction" in Methods |
| Recombinant DNA reagent | pSXY1035 (CP43-W353F) | This paper | | See "Plasmid construction" in Methods |
| Recombinant DNA reagent | pSXY1036 (CP43-W375A) | This paper | | See "Plasmid construction" in Methods |
| Recombinant DNA reagent | pSXY1037 (CP43-W375F) | This paper | | See "Plasmid construction" in Methods |
| Recombinant DNA reagent | pSXY1038 (CP43-W353A-W375A) | This paper | | See "Plasmid construction" in Methods |

*Appendix 1 Continued on next page*

*Appendix 1 Continued*

| Reagent type (species) or resource | Designation | Source or reference | Identifiers | Additional information |
|---|---|---|---|---|
| Recombinant DNA reagent | pSXY1039 (CP43-W353F-W375F) | This paper | | See "Plasmid construction" in Methods |
| Commercial assay or kit | *MagnosphereTM, MS 160/Tosyl* | MBL life sciences | J-MS-S160T | |
| Commercial assay or kit | *BlockmasterTM CE210* | MBL life sciences | J-CE210RAN | |
| Commercial assay or kit | *Pierce BCA Protein Assay Kit* | Thermo Fisher | Cat # 23225 | |
| Software, algorithm | Proteome Discoverer 2.4 | Thermo Fisher Scientific | | |
| Software, algorithm | ImageJ 1.53 k | https://imagej.nih.gov/ | | |
| Software, algorithm | NAMD 2.13 | https://www.ks.uiuc.edu/Research/namd/ | | |
| Software, algorithm | Jaguar 8.0 | Schrödinger, LLC. | | |
| Software, algorithm | Amber 14 | https://ambermd.org/ | | |
| Software, algorithm | Molecular Operating Environment (MOE), 2018.01 | Chemical Computing Group ULC. | | |
| Sequence-based reagent | On#1 | This paper | PCR primers | 5'-GGCTCGAGATGCATACGCGTGCTAGCCGAACGCCAGCAAGACGTAGCCC |
| Sequence-based reagent | On#2 | This paper | PCR primers | 5'-GGGAATTCACGCGTTCGAACATGAGCGCTTGTTTCGGCGTGG |
| Sequence-based reagent | On#3 | This paper | PCR primers | 5'-AGCTTAGATCTGTCGACGATATCACTAGTT |
| Sequence-based reagent | On#4 | This paper | PCR primers | 5'-CTAGAACTAGTGATATCGTCGACAGATCTA |
| Sequence-based reagent | On#5 | This paper | PCR primers | 5'-AAAAAAATTAACATATGACAGCGCTCGTTTTTGTGAGTGGATCACTTC |
| Sequence-based reagent | On#6 | This paper | PCR primers | 5'-AATTTTTTTGTCATATGTGTAATGTATTATAAAATTTATTTGCCCG |
| Sequence-based reagent | On#7 | This paper | PCR primers | 5'-TTAACATATGACAGCAATTTTAGAACGTCGTG |
| Sequence-based reagent | On#8 | This paper | PCR primers | 5'-CTCACAAAAACGAGCAGCTAGGCTAGAATTTTC |
| Sequence-based reagent | On#9 | This paper | PCR primers | 5'-CTCACAAAAACGAGCGAATAGGCTAGAATTTTC |
| Sequence-based reagent | On#10 | This paper | PCR primers | 5'-CACCTGTAGCTTGGCTGCTGATTACC |
| Sequence-based reagent | On#11 | This paper | PCR primers | 5'-AGGTTTATCAACTATGGCATTCAACTTAAACGG |

*Appendix 1 Continued on next page*

*Appendix 1 Continued*

| Reagent type (species) or resource | Designation | Source or reference | Identifiers | Additional information |
| --- | --- | --- | --- | --- |
| Sequence-based reagent | On#12 | This paper | PCR primers | 5'-GTTGATGATGTCTGCAGCAGTGTTTAGTACACGACCTTGTGAG |
| Sequence-based reagent | On#13 | This paper | PCR primers | 5'-GCAGACATCATCAACCGTGCTAACTTAG |
| Sequence-based reagent | On#14 | This paper | PCR primers | 5'-CTAAAATAAACCAGGTATGGTTAACCAGATTTATT |
| Sequence-based reagent | On#15 | This paper | PCR primers | 5'-GTTGATGATGTCTGCGAAAGTGTTTAGTACACGACCTTGTGAG |
| Sequence-based reagent | On#16 | This paper | PCR primers | 5'-GCCAACTGCCTATGGTAGCTATTAAG |
| Sequence-based reagent | On#17 | This paper | PCR primers | 5'-GAGCGGATAACAATTTCACACAGG |
| Sequence-based reagent | On#18 | This paper | PCR primers | 5'-CGCCAGGGTTTTCCCAGTCACGAC |
| Sequence-based reagent | On#19 | This paper | PCR primers | 5'-AAATGATTTCACCTGTTGGAGAACGCATTA |
| Sequence-based reagent | On#20 | This paper | PCR primers | 5'-AAATTAACGCAGTTAACTTCGTATCTCCAC |
| Sequence-based reagent | On#21 | This paper | PCR primers | 5'-AAATTAACGCAGTAAACTTCGTATCTCCAC |
| Sequence-based reagent | On#22 | This paper | PCR primers | 5'-TTAGTCTAAAGGACGCATTGAAAGAACTGG |
| Sequence-based reagent | On#23 | This paper | PCR primers | 5'-ACTCATGGTGGCAAGTTCGATAAACTTTAG |
| Sequence-based reagent | On#24 | This paper | PCR primers | 5'-GGGAGCTCCAGAAGAAATTGTGACAACATGGATC |
| Sequence-based reagent | On#25 | This paper | PCR primers | 5'-TTATTTGTGCTAAAAATTGCAGTTGGAAAG |
| Sequence-based reagent | On#26 | This paper | PCR primers | 5'-GGACCACGGAAGTCCCAGAAACGCATAG |
| Sequence-based reagent | On#27 | This paper | PCR primers | 5'-TGCAGAACGTCGTGCTGCTGAATACATGACTC |
| Sequence-based reagent | On#28 | This paper | PCR primers | 5'-AGCAGCACGACGTTCTTGCCAAGGTTG |
| Sequence-based reagent | On#29 | This paper | PCR primers | 5'-GGACTTCCGTGGTCCAGCTTTAGAACCTCTACGTGGTCCAAACGG |
| Sequence-based reagent | On#30 | This paper | PCR primers | 5'-GGACTTCCGTGGTCCATTCTTAGAACCTCTACGTGGTCCAAACGG |
| Sequence-based reagent | On#31 | This paper | PCR primers | 5'-GGACTTCCGTGGTCCATGGTTAGAACCTC |
| Sequence-based reagent | On#32 | This paper | PCR primers | 5'-AGCAGCACGACGTTCTTGAGCAGGTTGAATATCATTTTTAAGTTTG |
| Sequence-based reagent | On#33 | This paper | PCR primers | 5'-AGCAGCACGACGTTCTTGGAAAGGTTGAATATCATTTTTAAGTTTG |

