## [Editor Report · eLife assessment]

This study adds a **fundamental** new perspective to a long-standing question: What controls the repair of photosystem II (PSII), a key process in maintaining and optimizing photosynthesis? The work supports a role for chemical modification in the recognition and subsequent degradation of a key protein subunit of PSII by a bacterial-type protease, suggesting that tryptophan oxidation of components of the photosynthetic apparatus after high light stress plays a critical role in initiating the PSII repair system. The evidence supporting the authors' conclusions is **solid**.

---

## [Referee Report · Reviewer #1 (Public Review)]

This manuscript tried to answer a long-standing question in an important research topic. I read it with great interest. The quality of the science is high, and the text is clearly written. The conclusion is exciting. However, I feel that the phenotype of the transgenic line may be explained by an alternative idea. At least, the results should be more carefully discussed.

Specific comments:

1. Stability or activity (Fv/Fm) was not affected in PSII with the W14F mutation in D1. If W14F really represents the status of PSII with oxidized D1, what is the reason for the degradation of almost normal D1?

2. To focus on the PSII in which W14 is oxidized, this research depends on the W14F mutant lines. It is critical how exactly the W-to-F substitution mimics the oxidized W. The authors tried to show it in Figure 5. Because of the technical difficulty, it may be unfair to request more evidence. But the paper would be more convincing with the results directly monitoring the oxidized D1 to be recognized by FtsH.

3. Figure 3. If the F14 mimics the oxidized W14 and is sensed by FtsH, I would expect the degradation of D1 even under the growth light. The actual result suggests that W14F mutation partially modifies the structure of D1 under high light and this structural modification of D1 is sensed by FtsH. Namely, high light may induce another event which is recognized by FtsH. The W14F is just an enhancer.

---

## [Referee Report · Reviewer #2 (Public Review)]

In their manuscript, Kato et al investigate a key aspect of membrane protein quality control in plant photosynthesis. They study the turnover of plant photosystem II (PSII), a hetero-oligomeric membrane protein complex that undertakes the crucial light-driven water oxidation reaction in photosynthesis. The formidable water oxidation reaction makes PSII prone to photooxidative damage. PSII repair cycle is a protein repair pathway that replaces the photodamaged reaction center protein D1 with a new copy. The manuscript addresses an important question in PSII repair cycle - how is the damaged D1 protein recognized and selectively degraded by the membrane-bound ATP-dependent zinc metalloprotease FtsH in a processive manner? The authors show that oxidative post-translational modification (OPTM) of the D1 N-terminus is likely critical for the proper recognition and degradation of the damaged D1 by FtsH. Authors use a wide range of approaches and techniques to test their hypothesis that the singlet oxygen (1O2)-mediated oxidation of tryptophan 14 (W14) residue of D1 to N-formylkynurenine (NFK) facilitates the selective degradation of damaged D1. Overall, the authors propose an interesting new hypothesis for D1 degradation and their hypothesis is supported by most of the experimental data provided. The study certainly addresses an elusive aspect of PSII turnover and the data provided go some way in explaining the light-induced D1 turnover. However, some of the data are correlative and do not provide mechanistic insight. A rigorous demonstration of OPTM as a marker for D1 degradation is yet to be made in my opinion. Some strengths and weaknesses of the study are summarized below:

Strengths:

1. In support of their hypothesis, the authors find that FtsH mutants of Arabidopsis have increased OPTM, especially the formation of NFK at multiple Trp residues of D1 including the W14; a site-directed mutation of W14 to phenylalanine (W14F), mimicking NFK, results in accelerated D1 degradation in Chlamydomonas; accelerated D1 degradation of W14F mutant is mitigated in an ftsH1 mutant background of Chlamydomonas; and that the W14F mutation augmented the interaction between FtsH and the D1 substrate.

2. Authors raise an intriguing possibility that the OPTM disrupts the hydrogen bonding between W14 residue of D1 and the serine 25 (S25) of PsbI. According to the authors, this leads to an increased fluctuation of the D1 N-terminal tail, and as a consequence, recognition and binding of the photodamaged D1 by the protease. This is an interesting hypothesis and the authors provide some molecular dynamics simulation data in support of this. If this hypothesis is further supported, it represents a significant advancement.

3. The interdisciplinary experimental approach is certainly a strength of the study. The authors have successfully combined mass spectrometric analysis with several biochemical assays and molecular dynamics simulation. These, together with the generation of transplastomic algal cell lines, have enabled a clear test of the role of Trp oxidation in selective D1 degradation.

4. Trp oxidative modification as a degradation signal has precedent in chloroplasts. The authors cite the case of 1O2 sensor protein EXECUTER 1 (EX1), whose degradation by FtsH2, the same protease that degrades D1, requires prior oxidation of a Trp residue. The earlier observation of an attenuated degradation of a truncated D1 protein lacking the N-terminal tail is also consistent with authors' suggestion of the importance of the D1 N-terminus recognition by FtsH. It is also noteworthy that in light of the current study, D1 phosphorylation is unlikely to be a marker for degradation as posited by earlier studies.

Weaknesses:

1. The study lacks some data that would have made the conclusions more rigorous and convincing. It is unclear why the level of Trp oxidation was not analyzed in the Chlamydomonas ftsH 1-1 mutant as done for the var 2 mutant. Increased oxidation of W14 OPTM in Chlamydomonas ftsH 1-1 is a key prediction of the hypothesis. It is also unclear to me what is the rationale for showing D1-FtsH interaction data only for the double mutant but not for the single mutant (W14F). Why is the FtsH pulldown of D2 not statistically significant (p value = {less than or equal to}0.1). Wouldn't one expect FtsH pulls down the RC47 complex containing D1, D2, and RC47. Probing the RC47 level would have been useful in settling this. A key proposition of the authors' is that the hydrogen bonding between D1 W14 and S25 of PsbI is disrupted by the oxidative modification of W14. Can this hypothesis be further tested by replacing the S25 of PsbI with Ala, for example?

2. Although most of the work described is in vivo analysis, which is desirable, some in vitro degradation assays would have strengthened the conclusions. An in vitro degradation assay using the recombinant FtsH and a synthetic peptide encompassing D1 N-terminus with and without OPTM will test the enhanced D1 degradation that the authors predict. This will also help to discern the possibility that whether CP43 detachment alone is sufficient for D1 degradation as suggested for cyanobacteria.

3. The rationale for analyzing a single oxidative modification (W14) as a D1 degradation signal is unclear. D1 N-terminus is modified at multiple sites. Please see Mckenzie and Puthiyaveetil, bioRxiv May 04 2023. Also, why is modification by only 1O2 considered while superoxide and hydroxide radicals can equally damage D1?

4. The D1 degradation assay seems not repeatable for the W14F mutant. High light minus CAM results in Fig. 3 shows a statistically significant decrease in D1 levels for W14F at multiple time points but the same assay in Fig. 4a does not produce a statistically significant decrease at 90 min of incubation. Why is this? Accelerated D1 degradation in the Phe mutant under high light is key evidence that the authors cite in support of their hypothesis.

5. The description of results at times is not nuanced enough, for e.g. lines 116-117 state "The oxidation levels in Trp-14 and Trp-314 increased 1.8-fold and 1.4-fold in var2 compared to the wild type, respectively (Fig. 1c)" while an inspection of the figure reveals that modification at W314 is significant only for NFK and not for KYN and OIA. Likewise, the authors write that CP43 mutant W353F has no growth phenotype under high light but Figure S6 reveals otherwise. The slow growth of this mutant is in line with the earlier observation made by Anderson et al., 2002. In lines 162-163, the authors talk about unchanged electron transport in some site-directed mutants and cite Fig. 2c but this figure only shows chl fluorescence trace and nothing else.

6. The authors rightly discuss an alternate hypothesis that the simple disassembly of the monomeric core into RC47 and CP43 alone may be sufficient for selective D1 degradation as in cyanobacteria. This hypothesis cannot yet be ruled out completely given the lack of some in vitro degradation data as mentioned in point 2. Oxidative protein modification indeed drives the disassembly of the monomeric core (Mckenzie and Puthiyaveetil, bioRxiv May 04 2023).

---

## [Referee Report · Reviewer #3 (Public Review)]

Light energy drives photosynthesis. However, excessive light can damage (i.e., photo-damage) and thus inactivate the photosynthetic process. A major target site of photo-damage is photosystem II (PSII). In particular, one component of PSII, the reaction center protein, D1, is very suspectable to photo-damage, however, this protein is maintained efficiently by an elaborate multi-step PSII-D1 turnover/repair cycle. Two proteases, FtsH and Deg, are known to contribute to this process, respectively, by efficient degradation of photo-damaged D1 protein processively and endoproteolytically. In this manuscript, Kato et al., propose an additional step (an early step) in the D1 degradation/repair pathway. They propose that "Tryptophan oxidation" at the N-terminus of D1 may be one of the key oxidations in the PSII repair, leading to processive degradation of D1 by FtsH. Both, their data and arguments are very compelling.

The D1 protein repair/degradation pathway in its simplest form can be defined essentially by five steps: (1) migration of damaged PSII core complex to the stroma thylakoid, (2) partial PSII disassembly of the PSII core monomer, (3) access of protease degrading damaged D1, (4) concomitant D1 synthesis, and (5) reassembly of PSII into grana thylakoid. An enormous amount of work has already been done to define and characterize these various steps. Kato et al., in this manuscript, are proposing a very early yet novel critical step in D1 protein turnover in which Tryptophan(Trp) oxidation in PSII core proteins influences D1 degradation mediated by FtsH.

Using a variety of approaches, such as mass-spectrometry (Table 1), site-directed mutagenesis (Figures 2-4), D1 degradation assays (Figures 3, and 4), and simulation modeling (Figure 5), Kato et al., provide both strong evidence and reasonable arguments that an N-terminal Trp oxidation may be likely to be a 'key' oxidative post-translational modification (OPTM) that is involved in triggering D1 degradation and thus activating the PSII repair pathway. Consequently, from their accumulated data, the authors propose a scenario in which the unraveling of the N-terminal of the D1 protein facilitated by Trp oxidation plays a critical 'recognition' role in alerting the plant that the D1 protein is photo-damaged and thus to kick start the processive degradation pathway initiated possibly by FtsH. Coincidently, Forsman and Eaton-Rye (Biochemistry 2021, 60, 1, 53-63), while working with the thermophilic cyanobacterium, Thermosynechococcus vulcanus, showed that when the N-terminal DE-loop of the D1 protein is photo-damaged a disruption of the interaction between the PsbT subunit and D1 occurs which may serve as a signal for PSII to undergo repair following photodamage. While the activation of the processive degradation pathways in Chlamydomonas versus Thermosynechococcus vulcanus have significant mechanistic differences, it's interesting to note and speculate that the stability of the N-terminal of their respective D1 proteins seems to play a critical role in 'signaling' the PSII repair system to be activated and initiate repair. But it's complicated. For instance, significant Trp oxidation also occurs on the lumen side of other PSII subunits which may also play a significant role in activating the repair processes as well. Indeed, Kato et al.,( Photosynthesis Research volume 126, pages 409-416 (2015)) proposed a two-step model whereby the primary event is disruption of a Mn-cluster in PSII on the lumen side. A secondary event is damage to D1 caused by energy that is absorbed by chlorophyll. But models adapt, change, and get updated. And the data provided by Kato et al., in this manuscript, gives us a unique glimpse/snapshot into the importance of the stability of the N-terminal during photo-damage and its role in D1-turnover. For instance, the author's use site-directed mutagenesis of Trp residues undergoing OPTM in the D1 protein coupled with their D1 degradation assays (Figure 3 and 4), provides evidence that Trp oxidation (in particular the oxidation of Trp14) in coordination with FtsH results in the degradation of D1 protein. Indeed, their D1 degradation assays coupled with the use of a ftsh mutant provide further significant support that Trp14 oxidation and FtsH activity are strongly linked. But for FstH to degrade D1 protein it needs to gain access to photo-damaged D1. FtsH access to D1 is achieved by having CP43 partially dissociate from the PSII complex. Hence, the authors also addressed the possibility that Trp oxidation may also play a role in CP43 disassembly from the PSII complex thereby giving FtsH access to D1. Using a site-directed mutagenesis approach, they showed that Trp oxidation in CP43 appeared to have little impact on the PSII repair (Supplemental Figure S6). This result shows that D1-Trp14 oxidation appears to be playing a role in D1 turnover that occurs after CP43 disassembly from the PSII complex. Alternatively, the authors cannot exclude the possibility that D1-Trp14 oxidation in some way facilitates CP43 dissociation. Further investigation is needed on this point. However, D1-Trp14 oxidation is causing an internal disruption of the D1 protein possibly at the N-terminus of the protein. Consequently, the role of Trp14 oxidation in disrupting the stability of the N-terminal domain of the D1 protein was analyzed computationally. Using a molecular dynamics approach (Figure 5), the authors attempted to create a mechanistic model to explain why when D1 protein Trp14 undergoes oxidation the N-terminal domain of D1protein becomes unraveled. Specifically, the authors propose that the interaction between D1 protein Trp14 with PsbI Ser25 becomes disrupted upon oxidation of Trp14. Consequently, the authors concluded from their molecular dynamics simulation analysis that " the increased fluctuation of the first α-helix of D1 would give a chance to recognize the photo-damaged D1 by FtsH protease". Hence, the author's experimental and computational approaches employed here develop a compelling early-stage repair model that integrates (1) Trp14 oxidation, (2) FtsH activation and (3) D1- turnover being initiated at its N-terminal domain. However, a word of caution should be emphasized here. This model is just a snapshot of the very early stages of the D1 protein turnover process. The data presented here gives us just a small glimpse into the unique relationship between Trp oxidation of the D1 protein which may trigger significant N-terminal structural changes of the D1 protein that both signals and provides an opportunity for FstH to begin protease digestion of the D1 protein. However, the authors go to great lengths in their discussion section to not overstate solely the role of Trp14 oxidation in the complicated process of D1 turnover. The authors certainly recognize that there are a lot of moving parts involved in D1 turnover. And while Trp14 oxidation is the major focus of this paper, the authors show in Supplemental Fig S4 the structural positions of various additional oxidized Trp residues in the Thermosynecoccocus vulcans PSII core proteins. Indeed, this figure shows that the majority of oxidized Trps are located on the luminal side of PSII complex clustered around the oxygen-evolving complex. So, while oxidized Trp14 may be involved in the early stages of D1 turnover certainly oxidized Trps on the lumen side are also more than likely playing a role in D1 turnover as well. To untangle this complex process will require additional research.

Nevertheless, identifying and characterizing the role of oxidative modification of tryptophan (Trp) residues, in particular, Trp14, in the PSII core provides another critical step in an already intricate multi-step process of D1 protein turnover during photo-damage.

---

## [Author Response]

The following is the authors’ response to the original reviews.

**Reviewer #1 (Public Review):**
This manuscript tried to answer a long-standing question in an important research topic. I read it with great interest. The quality of the science is high, and the text is clearly written. The conclusion is exciting. However, I feel that the phenotype of the transgenic line may be explained by an alternative idea. At least, the results should be more carefully discussed.

We thank the reviewer #1 for his/her comments that helped to improve the manuscript. We have incorporated changes to reflect the suggestions provided by the reviewer. Here is a point-by-point response to the reviewer's specific and other minor comments.

Specific comments:1. Stability or activity (Fv/Fm) was not affected in PSII with the W14F mutation in D1. If W14F really represents the status of PSII with oxidized D1, what is the reason for the degradation of almost normal D1?

In this study, we used W14F mutation to mimic Trp-14 oxidation. The W14F mutant did not affect the stability and photosynthetic activity under normal growth conditions. However, the W14F mutant showed increased D1 degradation and reduced Fv/Fm values under high light. These results suggested that the W14F mutant has almost normal D1 protein stability under growth light conditions, as pointed out by the reviewer.

However, it should be noted that D1 protein in the W14F strain rapidly degraded under high light. In the discussion part, we mentioned the possibility that other OPTMs may have additive effects on D1 degradation. Synergistic effects such as different amino acid oxidations may cause D1 degradation, and among those oxidative damages, W14 oxidation would be a key signal for D1 degradation by FtsH.

1. To focus on the PSII in which W14 is oxidized, this research depends on the W14F mutant lines. It is critical how exactly the W-to-F substitution mimics the oxidized W. The authors tried to show it in Figure 5. Because of the technical difficulty, it may be unfair to request more evidence. But the paper would be more convincing with the results directly monitoring the oxidized D1 to be recognized by FtsH.

We agree that confirming the direct interaction of oxidized D1 protein with FtsH provides more robust evidence. However, since FtsH progressively degrades the trapped substrate, it would be quite a challenging attempt to capture that moment. There are also technical limitations to obtaining sufficient substrate using Co-IP to compare its oxidation state. We included your suggested point in the discussion part. Thank you for your valuable suggestion.

1. Figure 3. If the F14 mimics the oxidized W14 and is sensed by FtsH, I would expect the degradation of D1 even under the growth light. The actual result suggests that W14F mutation partially modifies the structure of D1 under high light and this structural modification of D1 is sensed by FtsH. Namely, high light may induce another event which is recognized by FtsH. The W14F is just an enhancer.

Our results indicated that W14 oxidation is one of the keys to D1 degradation. On the other hand, we agree with the possibility that the reviewer points out. There is the possibility that factors other than W14 may act synergistically to promote D1 degradation. High light triggered more D1 degradation in W14F, suggesting that unknown factor(s) may be required for D1 degradation, e.g., oxidative modification at other sites and/or conformational changes of PSII under the high light. However, the current data that we have cannot reveal. We have incorporated the reviewer's comment and discussed it in the discussion part.

**Reviewer #2 (Public Review):**
In their manuscript, Kato et al investigate a key aspect of membrane protein quality control in plant photosynthesis. They study the turnover of plant photosystem II (PSII), a hetero-oligomeric membrane protein complex that undertakes the crucial light-driven water oxidation reaction in photosynthesis. The formidable water oxidation reaction makes PSII prone to photooxidative damage. PSII repair cycle is a protein repair pathway that replaces the photodamaged reaction center protein D1 with a new copy. The manuscript addresses an important question in PSII repair cycle - how is the damaged D1 protein recognized and selectively degraded by the membrane-bound ATP-dependent zinc metalloprotease FtsH in a processive manner? The authors show that oxidative post-translational modification (OPTM) of the D1 N-terminus is likely critical for the proper recognition and degradation of the damaged D1 by FtsH. Authors use a wide range of approaches and techniques to test their hypothesis that the singlet oxygen (1O2)-mediated oxidation of tryptophan 14 (W14) residue of D1 to N-formylkynurenine (NFK) facilitates the selective degradation of damaged D1. Overall, the authors propose an interesting new hypothesis for D1 degradation and their hypothesis is supported by most of the experimental data provided. The study certainly addresses an elusive aspect of PSII turnover and the data provided go some way in explaining the light-induced D1 turnover. However, some of the data are correlative and do not provide mechanistic insight. A rigorous demonstration of OPTM as a marker for D1 degradation is yet to be made in my opinion. Some strengths and weaknesses of the study are summarized below:

We thank reviewer #2 for his/her comments that helped to improve the manuscript. We have incorporated changes to reflect the suggestions pointed out as weaknesses by reviewer #2. Other minor comments were also answered in a point-by-point response.

Strengths:1. In support of their hypothesis, the authors find that FtsH mutants of Arabidopsis have increased OPTM, especially the formation of NFK at multiple Trp residues of D1 including the W14; a site-directed mutation of W14 to phenylalanine (W14F), mimicking NFK, results in accelerated D1 degradation in Chlamydomonas; accelerated D1 degradation of W14F mutant is mitigated in an ftsH1 mutant background of Chlamydomonas; and that the W14F mutation augmented the interaction between FtsH and the D1 substrate.1. Authors raise an intriguing possibility that the OPTM disrupts the hydrogen bonding between W14 residue of D1 and the serine 25 (S25) of PsbI. According to the authors, this leads to an increased fluctuation of the D1 N-terminal tail, and as a consequence, recognition and binding of the photodamaged D1 by the protease. This is an interesting hypothesis and the authors provide some molecular dynamics simulation data in support of this. If this hypothesis is further supported, it represents a significant advancement.1. The interdisciplinary experimental approach is certainly a strength of the study. The authors have successfully combined mass spectrometric analysis with several biochemical assays and molecular dynamics simulation. These, together with the generation of transplastomic algal cell lines, have enabled a clear test of the role of Trp oxidation in selective D1 degradation.1. Trp oxidative modification as a degradation signal has precedent in chloroplasts. The authors cite the case of 1O2 sensor protein EXECUTER 1 (EX1), whose degradation by FtsH2, the same protease that degrades D1, requires prior oxidation of a Trp residue. The earlier observation of an attenuated degradation of a truncated D1 protein lacking the N-terminal tail is also consistent with authors' suggestion of the importance of the D1 N-terminus recognition by FtsH. It is also noteworthy that in light of the current study, D1 phosphorylation is unlikely to be a marker for degradation as posited by earlier studies.Weaknesses:1. The study lacks some data that would have made the conclusions more rigorous and convincing. It is unclear why the level of Trp oxidation was not analyzed in the Chlamydomonas ftsH 1-1 mutant as done for the var 2 mutant. Increased oxidation of W14 OPTM in Chlamydomonas ftsH 1-1 is a key prediction of the hypothesis.

We thank the reviewer for this valuable comment. We agree with the reviewer that the analysis of oxidized Trp level will reinforce the importance of Trp oxidation in the N-terminal of D1.In our preliminary experiment, we observed a trend toward increase of the kynurenine in Trp-14 in Chlamydomonas ftsH1-1 strain. However, we found large errors, and we could not conclude that this trend is significant. A possible reason for the large error was that the signal intensity of oxidized Trp was insufficient for quantification in a series of Chlamydomonas experiment. In addition, the fact that the amount of D1 in each culture was not stable also might be one reason. On the other hand, we keep note of a previous result that more fragmentation of D1 protein was observed in the Chlamydomonas ftsH1-1 mutant compared to that in Arabidopsis (Malnoë et al., Plant Cell 2014). This result suggests that an alternative D1 degradation pathway involving other proteases is more active in the Chlamydomonas ftsH1-1 mutant than in Arabidopsis var2 mutant. Furthermore, the Chlamydomonas ftsH1-1 mutant, caused by an amino acid substitution, still has a significant FtsH1/FtsH2 heterohexamer, and the level of FtsH1 and FtsH2 proteins increases significantly under high light irradiation. This is a significant difference from the Arabidopsis var2 mutant lacking FtsH2 subunit and showed reduced protein accumulation. These factors may explain to the lower detection levels of oxidized Trp in Chlamydomonas. We believe that improved sensitivity for detection of oxidized Trp peptides and more sophisticated experimental systems could solve this issue in the future.

It is also unclear to me what is the rationale for showing D1-FtsH interaction data only for the double mutant but not for the single mutant (W14F).

We thank the reviewer for the comment. As suggested by the reviewer, the analysis of the mutant crossing ftsH and W14F single mutation will provide more convincing evidence. Fig.3 showed that the photosensitivity in both W14F and W14FW317F was caused by the enhanced D1 degradation observed, which was due to the W14F mutation. Therefore, we crossed the ftsH mutant with W14FW317F, which has a more severe phenotype, to confirm whether FtsH is involved in this D1 degradation.

Why is the FtsH pulldown of D2 not statistically significant (p value = {less than or equal to}0.1). Wouldn't one expect FtsH pulls down the RC47 complex containing D1, D2, and RC47. Probing the RC47 level would have been useful in settling this.

For the immunoblot result of D2 and its statistical analysis, we answered in the following comment; No.2 in the reviewer's comment in Recommendations For The Authors.

We agree with the reviewer's suggestion that further immunoblot analysis for CP47 protein would help our understanding of FtsH and RC47 interaction. Indeed, we attempted the immunoblot analysis of CP47 after the FtsH Co-IP experiment. However, the detection of CP43 protein was not sensitive enough. This reason may be due to the lower titer of the CP47 antibody compared to the D1 and D2 antibodies.

A key proposition of the authors' is that the hydrogen bonding between D1 W14 and S25 of PsbI is disrupted by the oxidative modification of W14. Can this hypothesis be further tested by replacing the S25 of PsbI with Ala, for example?

It is an interesting question whether amino acid substitution in PsbI-S25 affects the stability of D1-N-term and its degradation by FtsH. We would like to analyze the possibility in the future. We thank the reviewer for this helpful suggestion.

1. Although most of the work described is in vivo analysis, which is desirable, some in vitro degradation assays would have strengthened the conclusions. An in vitro degradation assay using the recombinant FtsH and a synthetic peptide encompassing D1 N-terminus with and without OPTM will test the enhanced D1 degradation that the authors predict. This will also help to discern the possibility that whether CP43 detachment alone is sufficient for D1 degradation as suggested for cyanobacteria.

In vitro experimental systems are interesting. However, FtsH is known to function as a hexamer, which has not yet been successfully reconstituted in vitro. Therefore, it would not be easy to perform an in vitro experimental system using the N-terminal synthetic peptide of D1 as a substrate. Thank you for your valuable suggestions.

1. The rationale for analyzing a single oxidative modification (W14) as a D1 degradation signal is unclear. D1 N-terminus is modified at multiple sites. Please see Mckenzie and Puthiyaveetil, bioRxiv May 04 2023. Also, why is modification by only 1O2 considered while superoxide and hydroxide radicals can equally damage D1?

We agree with the possibility that oxidative modifications in other amino acids are also involved in the D1 degradation, as pointed out by the reviewer. We also thank the reviewer for pointing us to the interesting article of Mckenzie and Puthiyaveetil et al. that showed additional oxidations occurred in the D1-Nterminus, which we had yet to be aware of when we submitted our manuscript. It will be interesting to see how these amino acid oxidations work with W14 oxidation on D1 degradation in the future. The oxidation of Trp by 1O2 can serve as a substrate for FtsH, as in the case of EX1, so we focused on the analysis of Trp oxidation. Single oxygen is believed to be the potential reactive species of Trp oxidation. However, the detected oxidative modifications in this study were not exactly sure depended on singlet oxygen. Thus, we changed several sentences that mention tryptophan oxidation by single oxygen.

1. The D1 degradation assay seems not repeatable for the W14F mutant. High light minus CAM results in Fig. 3 shows a statistically significant decrease in D1 levels for W14F at multiple time points but the same assay in Fig. 4a does not produce a statistically significant decrease at 90 min of incubation. Why is this? Accelerated D1 degradation in the Phe mutant under high light is key evidence that the authors cite in support of their hypothesis.

In Fig. 4a, the p-value comparing the D1 level at 90 min between control and W14F was 0.1075. This value is slightly larger than 0.1. The result that one of the control experiments showed a decrease in D1 level relative to 0 h might cause this value. Given that the D1 level of the remaining three of the four replicates was unchanged in the control experiments, it can be considered an outlier. We believe the results do not affect our hypothesis that the earlier D1 degradation is occurred in W14F.

1. The description of results at times is not nuanced enough, for e.g. lines 116-117 state "The oxidation levels in Trp-14 and Trp-314 increased 1.8-fold and 1.4-fold in var2 compared to the wild type, respectively (Fig. 1c)" while an inspection of the figure reveals that modification at W314 is significant only for NFK and not for KYN and OIA.

In this sentence, we described the result that is compared with the oxidized peptide levels calculated from all Trp-oxidized derivatives. However, as pointed out by the reviewer, it was not correct to explain the result of Fig.1C. We corrected the sentence following the reviewer's suggestion as below;“The levels of Trp-oxidized derivatives, OIA, NFK, and KYN in Trp-14 and the level of KYN in Trp-314 were significantly increased in var2 compared to the wild type, respectively (Fig. 1c). "

Likewise, the authors write that CP43 mutant W353F has no growth phenotype under high light but Figure S6 reveals otherwise. The slow growth of this mutant is in line with the earlier observation made by Anderson et al., 2002.

As pointed out by the reviewer, the growth of W353F seems to be a little slow under HL. We have changed our description of the result part. However, we still conclude that CP43 had little impact on the PSII repair, because the impaired growth in W353F is not as severe as those in W14F and W14F/W317F under HL

In lines 162-163, the authors talk about unchanged electron transport in some site-directed mutants and cite Fig. 2c but this figure only shows chl fluorescence trace and nothing else.

We agreed with the reviewer's suggestion and changed the sentence. In this study, we did not perform detailed photosynthetic analysis. Based on the analysis of phototrophic growth, oxygen-evolving activity, and Chl fluorescence, we concluded that overall photosynthetic activity was not a significant difference in the mutants.

1. The authors rightly discuss an alternate hypothesis that the simple disassembly of the monomeric core into RC47 and CP43 alone may be sufficient for selective D1 degradation as in cyanobacteria. This hypothesis cannot yet be ruled out completely given the lack of some in vitro degradation data as mentioned in point 2. Oxidative protein modification indeed drives the disassembly of the monomeric core (Mckenzie and Puthiyaveetil, bioRxiv May 04 2023).

Thanks for your suggestion. We added a discussion of PSII disassembly by ROS-induced oxidation to the discussion part, and the reference is added.

**Reviewer #3 (Public Review):**
Light energy drives photosynthesis. However, excessive light can damage (i.e., photo-damage) and thus inactivate the photosynthetic process. A major target site of photo-damage is photosystem II (PSII). In particular, one component of PSII, the reaction center protein, D1, is very suspectable to photo-damage, however, this protein is maintained efficiently by an elaborate multi-step PSII-D1 turnover/repair cycle. Two proteases, FtsH and Deg, are known to contribute to this process, respectively, by efficient degradation of photo-damaged D1 protein processively and endoproteolytically. In this manuscript, Kato et al., propose an additional step (an early step) in the D1 degradation/repair pathway. They propose that "Tryptophan oxidation" at the N-terminus of D1 may be one of the key oxidations in the PSII repair, leading to processive degradation of D1 by FtsH. Both, their data and arguments are very compelling.The D1 protein repair/degradation pathway in its simplest form can be defined essentially by five steps: (1) migration of damaged PSII core complex to the stroma thylakoid, (2) partial PSII disassembly of the PSII core monomer, (3) access of protease degrading damaged D1, (4) concomitant D1 synthesis, and (5) reassembly of PSII into grana thylakoid. An enormous amount of work has already been done to define and characterize these various steps. Kato et al., in this manuscript, are proposing a very early yet novel critical step in D1 protein turnover in which Tryptophan(Trp) oxidation in PSII core proteins influences D1 degradation mediated by FtsH.Using a variety of approaches, such as mass-spectrometry (Table 1), site-directed mutagenesis (Figures 2-4), D1 degradation assays (Figures 3, and 4), and simulation modeling (Figure 5), Kato et al., provide both strong evidence and reasonable arguments that an N-terminal Trp oxidation may be likely to be a 'key' oxidative post-translational modification (OPTM) that is involved in triggering D1 degradation and thus activating the PSII repair pathway. Consequently, from their accumulated data, the authors propose a scenario in which the unraveling of the N-terminal of the D1 protein facilitated by Trp oxidation plays a critical 'recognition' role in alerting the plant that the D1 protein is photo-damaged and thus to kick start the processive degradation pathway initiated possibly by FtsH. Coincidently, Forsman and Eaton-Rye (Biochemistry 2021, 60, 1, 53-63), while working with the thermophilic cyanobacterium, Thermosynechococcus vulcanus, showed that when the N-terminal DE-loop of the D1 protein is photo-damaged that occurs which may serve as a signal for PSII to undergo repair following photodamage. While the activation of the processive degradation pathways in Chlamydomonas versus Thermosynechococcus vulcanus have significant mechanistic differences, it's interesting to note and speculate that the stability of the N-terminal of their respective D1 proteins seems to play a critical role in 'signaling' the PSII repair system to be activated and initiate repair. But it's complicated. For instance, significant Trp oxidation also occurs on the lumen side of other PSII subunits which may also play a significant role in activating the repair processes as well. Indeed, Kato et al.,( Photosynthesis Research volume 126, pages 409-416 (2015)) proposed a two-step model whereby the primary event is disruption of a Mn-cluster in PSII on the lumen side.A secondary event is damage to D1 caused by energy that is absorbed by chlorophyll. But models adapt, change, and get updated. And the data provided by Kato et al., in this manuscript, gives us a unique glimpse/snapshot into the importance of the stability of the N-terminal during photo-damage and its role in D1-turnover. For instance, the author's use site-directed mutagenesis of Trp residues undergoing OPTM in the D1 protein coupled with their D1 degradation assays (Figure 3 and 4), provides evidence that Trp oxidation (in particular the oxidation of Trp14) in coordination with FtsH results in the degradation of D1 protein. Indeed, their D1 degradation assays coupled with the use of a ftsh mutant provide further significant support that Trp14 oxidation and FtsH activity are strongly linked. But for FstH to degrade D1 protein it needs to gain access to photo-damaged D1. FtsH access to D1 is achieved by having CP43 partially dissociate from the PSII complex. Hence, the authors also addressed the possibility that Trp oxidation may also play a role in CP43 disassembly from the PSII complex thereby giving FtsH access to D1. Using a site-directed mutagenesis approach, they showed that Trp oxidation in CP43 appeared to have little impact on the PSII repair (Supplemental Figure S6). This result shows that D1-Trp14 oxidation appears to be playing a role in D1 turnover that occurs after CP43 disassembly from the PSII complex. Alternatively, the authors cannot exclude the possibility that D1-Trp14 oxidation in some way facilitates CP43 dissociation. Further investigation is needed on this point. However, D1-Trp14 oxidation is causing an internal disruption of the D1 protein possibly at the N-terminus of the protein. Consequently, the role of Trp14 oxidation in disrupting the stability of the N-terminal domain of the D1 protein was analyzed computationally. Using a molecular dynamics approach (Figure 5), the authors attempted to create a mechanistic model to explain why when D1 protein Trp14 undergoes oxidation the N-terminal domain of D1protein becomes unraveled. Specifically, the authors propose that the interaction between D1 protein Trp14 with PsbI Ser25 becomes disrupted upon oxidation of Trp14. Consequently, the authors concluded from their molecular dynamics simulation analysis that " the increased fluctuation of the first α-helix of D1 would give a chance to recognize the photo-damaged D1 by FtsH protease". Hence, the author's experimental and computational approaches employed here develop a compelling early-stage repair model that integrates (1) Trp14 oxidation, (2) FtsH activation and (3) D1- turnover being initiated at its N-terminal domain. However, a word of caution should be emphasized here. This model is just a snapshot of the very early stages of the D1 protein turnover process. The data presented here gives us just a small glimpse into the unique relationship between Trp oxidation of the D1 protein which may trigger significant N-terminal structural changes of the D1 protein that both signals and provides an opportunity for FstH to begin protease digestion of the D1 protein.However, the authors go to great lengths in their discussion section to not overstate solely the role of Trp14 oxidation in the complicated process of D1 turnover. The authors certainly recognize that there are a lot of moving parts involved in D1 turnover. And while Trp14 oxidation is the major focus of this paper, the authors show in Supplemental Fig S4 the structural positions of various additional oxidized Trp residues in the Thermosynecoccocus vulcans PSII core proteins. Indeed, this figure shows that the majority of oxidized Trps are located on the luminal side of PSII complex clustered around the oxygen-evolving complex. So, while oxidized Trp14 may be involved in the early stages of D1 turnover certainly oxidized Trps on the lumen side are also more than likely playing a role in D1 turnover as well. To untangle this complex process will require additional research.Nevertheless, identifying and characterizing the role of oxidative modification of tryptophan (Trp) residues, in particular, Trp14, in the PSII core provides another critical step in an already intricate multi-step process of D1 protein turnover during photo-damage.

We thank reviewer #3 for all the helpful comments and their supportive review of the manuscript.

We thank the reviewer for raising this interesting study that ROS might disrupt the interaction between the PsbT and D1 in Thermosynechococcus vulcanus. The stroma-exposed DE-loop of D1 is one of the possible cleavage sites by Deg protease. Because the D1 cleavage by Deg facilitates the effective D1 degradation by FtsH under high-light conditions, it is interesting to elucidate Deg and FtsH cooperative D1 degradation further. We added this discussion in the manuscript.Other minor comments were also answered in a point-by-point response.

**Reviewer #1 (Recommendations For The Authors):**
Other minor points1. L227. How do you eliminate the possibility of reduced stability under high light?

D1 synthesis under HL as pointed out by the reviewer was not tested in this study. Therefore, we can not rule out the possibility of a reduced D1 synthesis rate under HL in the mutant. However, the rate of D1 turnover(coordinated degradation and synthesis) is increased under HL. Since the pulse-labeling experiment is affected D1 degradation as well as D1 synthesis, even if there is a difference in the rate of D1 synthesis under HL, we can not clearly distinguish whether the cause of reduced labeling is the increased D1 degradation seen in the W14F mutant or the delay in D1 synthesis. We thank the reviewer for this valuable comment.

1. Ls25-26. It would be quite rare that P680 directly absorbs light energy.

We changed the sentence.

1. L28. intrinsic antenna? Is this commonly used? core antenna?

Corrected to “core antenna”

1. Ls4143. Because the process is described as step (iii), it is curious to mention it again as other critical steps.

We removed the sentence.

1. L75. Is it correct? Do you mean damage is caused by inhibition?

We changed the sentence to “…the disorder of photosynthesis…”

1. Figure 1c. +4, +16 and +32 should be explained in the legend.

We added the explanation in the legend.

1. Supplementary Figures S1 and S2. Title. Is it true that oxidation depends on singlet oxygen? This is a question. If it is not experimentally proved, modify the expression.

In general, singlet oxygen (1O2) is believed to contribute in vivo oxidation of Trp. However, as suggested, these detected oxidative modifications were not exactly sure depends on singlet oxygen. Thus, we changed the title of Fig S1 and S2.

1. Figure 3. Correct errors in + or - in the Figure.

Corrected

1. L328. Cyc > Cys.

Corrected

**Reviewer #2 (Recommendations For The Authors):**
1. A few suggestions on typos and style:Lines 2-3, please rephrase the sentence. The meaning is unclear.

rephased the sentence to “Photosynthesis is one of the most …”

Lines 28-29, "Despite its orchestrated coordination...". Tautology.

We changed the sentence.

Line 31, "...one, known as the PSII repair...". Please rewrite.

We followed the reviewer suggestion and changed the sentence to “…synthesized one in the PSII repair.”

Line 49, "Their family proteins...". Rephrase.

Rephrased the words.

Lines 64-66, please rewrite. I am not sure what the authors imply here. Are they talking about FtsH turnover or regulation of FtsH at the protein or gene level?

FtsH itself is also degraded under high-light stress. To compensate for this, ftsH gene expression is upregulated and contributes to the proper FtsH level in thylakoid membranes. We rewrote the sentence as follows “increased turnover of FtsH is crucial for their function under high-light stress. That is compensated by upregulated FtsH gene expression”.

Line 68, "...to dislocate their substrates..."

We changed the sentence to “to pull their substrates and push them into the protease chamber by ATPase activity”

Line 86, N-formylkymurenine => N-formylkynurenine

Corrected

Lines 111-112, "Consistent with previous results...". Please specify which studies are being referred to and cite them if relevant.

We added references.

Line 114, "...in extracts Arabidopsis..." => "...in extracts of Arabidopsis...".

Corrected

Line 171, "influences in high-light sensitivity." Please rephrase.

We rephrased the sentence.

Line 192, Fv/Fm. "v" and "m" should be subscripts.

Corrected

Line 210, "...encounters...". Unclear meaning.

We rephrased the sentence.

Line 358, hyphen usage. "fine-tuned". This sentence should be rewritten to make the role of phosphorylation clear. "Fine-tuning" is vague.

We changed the sentence to “…spatiotemporal regulation of D1 degradation”

Fig. 6 legend, luminal => lumenal

Changed to luminal

1. The statistical notation used for some results is confusing. In Fig. 6b, "*" stands for p = {less than or equal to}0.1 while in fig. 4 it denotes p = {less than or equal to}0.05. If this is not a typo, this usage deviates from the standard one. How is a D2 change in Fig. 6b significant given its p value of {less than or equal to}0.1? The Fig. 6b key for D2 does not correspond with the histogram pattern.

Thank you for your comments and suggestions. The asterisk in the Figure 6b is not a typo. We revised p value sign for less than 0.05 with a single asterisk to avoid confusion. While the case of p value in less than 0.1, we applied section sign “§” instead of the single asterisk sign to avoid confusion. Generally accepted p value to indicate statistically difference is less than 0.05. We found that D1 was p = 0.03322 and D2 was p = 0.07418. As we suspect these p value differences, the results for D2 protein detection were somewhat fluctuating while not in D1 protein detection as you commented. Still the reason of the fluctuating result of D2 signal intensity is not clear yet, we found the p value was between 0.05 and 0.10. We also rewrite the description in the corresponding result part.

1. There are no error bars in Fig. 5d while the error bars in Fig. 5e show that there are no significant differences between Cβ distances of W14F and W14ox with WT contrary to the authors' assertion in the text (lines 254-255).

The reason that there are no error bars in Fig. 5d. is because the fluctuation value in Fig. 5d was calculated from the entire trajectory (i.e., all snapshots) of the MD simulation. In contrast, the Cβ-Cβ distance value can be obtained at each individual snapshot of the simulation. Thus, Fig. 5e shows the averaged distances with the standard deviations (the error bars) over all these snapshots.To prevent any confusion for the reader, we have explicitly described “averaged Cβ-Cβ distance” and added an explanation of the error bars in the caption of Fig. 5e. It is important to note that our focus in the text (lines 254-255) was not on comparing the Cβ-Cβ distance of W14F with that of W14ox but the distance of W14F or W14ox with that of WT.

1. Figure 3 legends and figure labels do not correspond. Fig. 3b should be labeled as High light - Chloramphenicol and likewise, fig 3c should read growth light + Chloramphenicol to be consistent with the legend.

Corrected

1. How are OPTM levels of D1 Trp residues normalized? Is it against unmodified peptides or total proteins?

Oxidation levels of three oxidative variants of Trp in Trp14 and Trp317 containing peptides were obtained by label-free MS analysis. Fig.1 shows the intensity values of oxidized variants of Trp14 and Trp317. In this analysis, the levels of unoxidized peptides were not significantly changed between var2 and WT.

1. Fig. 1a cartoon might need work. It looks like the oxygen atom in OIA is misplaced.

Corrected

**Reviewer #3 (Recommendations For The Authors):**
In regard to Table 1, the sequence of the mass spectra fragment listed for Trp14 (i.e., ENSSL(W*)AR ) in Table 1 is different from the sequence of the mass spectra fragment of Trp14 shown in Supplemental Figure S1 (i.e., ESESLWGR). Likewise, the sequence of the mass spectra fragment listed for Trp317 (i.e., VLNT(W*)ADIINR ) in Table 1 is different from the sequence of the mass spectra fragment of Trp14 shown in Supplemental Figure S2 (i.e., VINTWADIINR). This discrepancy, I think can be simply explained.

Table 1 shows the newly detected peptide of Trp oxidation in PSII core protein in Chlamydomonas. On the other hand, Figures S1 and S2 are the results of MS analysis used for the level of Trp oxidation analysis in Arabidopsis var2 mutant, as shown in Fig. 1C. To avoid confusion, we added in the supplemental figure title that it was detected in Arabidopsis.

Labeling: In Figure 3, the figure legend states that b, high-light in the absence of CAM; but panel b, shows +CAM conditions. I think this labeling is incorrect and needs to be -CAM. Likewise, the figure legend states that c, growth-light in the presence of CAM. I think this labeling is incorrect and needs to be +CAM.

Corrected

This reviewer has a few comments/suggestions on the presentation of the sequence alignments showing the various positions of oxidized Trps within the D1(Figure 1), D2 and CP43 (Supplemental Figure S3) and CP47 (Supplemental Figure S3):The authors should consider highlighting in red all the various Trps shown in Table 1 with the corresponding alignments shown in Figure 1 for D1 protein and corresponding alignments in Supplemental Figure S3 (for D2 and CP43) and Supplemental Figure S3 continued (For CP47). Highlighting the locations of oxidized Trps across various species is very informative but as presented here the red labeling somewhat is haphazard, confusing and thus these figures lose some of their impact factor. For instance, in Supplementary Fig. S4, the reader can visualize the structural positions of oxidized Trp residues in the Thermosynecoccocus vulcanus PSII core proteins. When one then looks at the various alignments presented by the authors, one can see that other species have a similar arrangement of oxidized Trp residues as well. Consequently, when you now collectively look at the data presented in Table 1, Figure 1, Supplemental Figure S3 and Supplemental Figure S4, a picture emerges that illustrates how common the phenomenon of overall Trp oxidation is and more specifically how oxidized Trp14 across species is playing a similar role in possibly activating D1 turnover. I think these Figures, if presented in a more comprehensive and unified fashion, will really add to the paper.

Thank you for your suggestion. In this study, we tried to show the identified oxidized Trp by the MS-MS analysis, the residue conservation in the sequences, and its position in the structure. Since we have to show a lot of information, combining them into one figure is difficult. We hope you understand the reason for this.